# MamF-like proteins are distant Tic20 homologs involved in organelle assembly in bacteria

Anja Paulus [1], Frederik Ahrens[1], Annika Schraut[1], Hannah Hofmann[1], Tim Schiller [1], Thomas Sura [2], Dörte Becher [2] & René Uebe [1] ✉

Organelle-specific protein translocation systems are essential for organelle biogenesis and maintenance in eukaryotes but thought to be absent from prokaryotic organelles. Here, we demonstrate that MamF-like proteins are crucial for the formation and functionality of bacterial magnetosome organelles. Deletion of *mamF*-like genes in the Alphaproteobacterium *Magnetospirillum gryphiswaldense* results in severe defects in organelle positioning, biomineralization, and magnetic navigation. These phenotypic defects result from the disrupted targeting of a subset of magnetosomal proteins that contain C-terminal glycine-rich integral membrane domains. Phylogenetic analyses reveal an ancient evolutionary link between MamF-like proteins and plastidial Tic20. Our findings redefine the molecular roles of MamF-like proteins and suggest that organelle-specific protein targeting systems also play a role in bacterial organelle formation.

Eukaryotes possess complex preprotein translocases of chloroplasts or mitochondria (Tic/Toc or Tim/Tom) to ensure biogenesis and maintenance of these organelles. Strikingly, both translocases have a mixed phylogenetic origin and likely evolved by combining new eukaryotic proteins with bacterial subunits of endosymbiotic origin[1–4]. However, no organelle-specific protein translocases equivalent to the eukaryotic machineries have been identified among bacteria yet[5]. Hence, bacterial organelle biogenesis must proceed through distinct mechanisms.

Magnetosomes, unique magnetic organelles[6], for example, emerge from the cell membrane by invagination[7]. It has been hypothesized that in *Magnetospirillum gryphiswaldense* MSR-1 and related Alphaproteobacteria, this process is induced by the assembly of magnetosome-specific protein complexes within the cytoplasmic membrane that initiate magnetosome membrane (MM) formation through a molecular crowding-like membrane-bending mechanism[8]. Thereby, magnetosome-specific proteins, which are encoded within a compact genomic magnetosome island (MAI), supposedly become integrated into the MM during its invagination. Some magnetosome proteins, however, follow different targeting routes. Mms6 and MamD

(also known as Mms7), for example, are translocated into preexisting magnetosomes in a folded state upon induction of magnetite biomineralization[9,10]. Nevertheless, the precise molecular mechanisms governing magnetosome assembly and protein targeting remained largely elusive.

Immediately following MM formation, the embedded magnetosome proteins facilitate subsequent steps of magnetosome biogenesis like iron uptake into the magnetosome lumen, initiation of magnetite ($Fe_3O_4$) biomineralization and magnetite crystal growth[11]. The small integral protein MamF and its paralogue MmsF, for example, are thought to promote magnetite crystal growth by direct interaction with the magnetite crystal surface or metal ions via a cluster of lumen-residing conserved acidic amino acids[12–14]. Concurrent with the biogenesis of mature magnetosomes, a dynamic magnetosome-specific cytoskeletal network orchestrates the assembly of approximately 30 magnetosomes into a linear chain. This so-called "magnetoskeleton" is composed of the MM-adaptor protein MamJ[15], which mediates the binding of magnetosomes to the soluble actin-like protein MamK[7] and the cell membrane-bound scaffold MamY[16]. The chain-like arrangement of magnetosomes creates a cellular magnetic dipole, formed by

[1]Department of Microbiology, University of Bayreuth, Bayreuth, Germany. [2]Microbial Proteomics, Institute of Microbiology, University of Greifswald, Greifswald, Germany. ✉e-mail: rene.uebe@uni-bayreuth.de

the collective dipole moments of individual magnetosomes. This dipole enables geomagnetic navigation, which is thought to aid magnetotactic bacteria to locate preferred anoxic or microoxic zones within their stratified aquatic habitats[11].

Here, we integrate comprehensive bioinformatic analyses with molecular, cell biological, quantitative proteomic, and biochemical studies to elucidate the mechanisms underlying magnetosome protein targeting and assembly. In combination with the systematic evaluation of key protein functions that are directly reflected by magnetism-related phenotypes, we demonstrate that magnetosomal MamF-like proteins (MFPs) mediate organelle-specific protein targeting in MSR-1 and are distantly related to the plastidal preprotein translocase core subunit Tic20.

## Results

### MFPs are members of a common Tic20/HOTT superfamily

Within a previous study, we discovered a ~ 10 kb region within the MSR-1 genome that encodes several hitherto unrecognized magnetosome proteins[17]. One of these proteins, MmxF, is closely related to the crystal size-regulating magnetosome proteins MamF and MmsF (hereafter collectively termed MamF-like proteins, MFPs) (Supplementary Fig. 1a)[12–14]. Contrary to earlier studies that failed to detect any homologous proteins outside of magnetotactic bacteria, we identified significant homologies between MFPs and the DUF4870 (hereafter termed HOTT for Homologs of Tic Twenty) as well as the Tic20 protein families (Fig. 1a) using HHpred, a highly sensitive Hidden Markov Model-based homology detection tool[18]. Interestingly, while the widely distributed HOTT family encompasses only uncharacterized proteins, the Tic20 family, among uncharacterized cyanobacterial representatives, also comprises Tic20 from chloroplasts (Supplementary Fig. 1b, c). In these photosynthetic organelles, Tic20 acts as a core component of the protein-translocating Tic complex at the inner envelope membrane[19–22]. The homologies identified by HHPred are further supported by cluster analysis of sequences (CLANS), which relies on all-against-all BLAST search-derived sequence similarity scores[23]. Here, the Tic20 family forms a widely separated but connected cluster to the HOTT family, in which MFPs seem to represent a discrete subfamily (Fig. 1b). Similar results were obtained by phylogenetic analyses where seven subfamilies including MFPs form distinct branches within the HOTT family, which itself is separated from the Tic20 family clade by a long branch (Supplementary Fig. 1d). Remarkably, despite only remote sequence homologies, multiple sequence alignments of HOTT and Tic20 family proteins, and structural superimposition using distance matrix alignment (DALI) analyses[24] revealed similar architectures, including a reentrant-like integral membrane helix (IMH) that is followed by a conserved charged loop and two additional integral membrane helices (Fig. 1c, d).

Collectively, our data show that the HOTT (including the MFPs) and Tic20 families are distantly related and constitute a common protein superfamily.

### MFPs are essential for magnetosome chain formation and magnetotaxis

The remote homology to Tic20 suggests that MFPs, besides regulating biomineralization, may also play a role during magnetosome assembly. To explore this hypothesis, we first analyzed transmission electron micrographs (TEM) of MSR-1 mutants carrying deletions of *mamF*-like genes in all possible combinations, including the MFP-free mutant ΔF3 (Δ*mamF*Δ*mmsF*Δ*mmxF*) (Fig. 2 and Supplementary Fig. 2). These analyses revealed that, in agreement with known MFP deletion phenotypes[12,13], the average magnetite crystal size gradually decreased from the WT, the single and double deletion mutants to the ΔF3 strain while crystal numbers per cell were only slightly reduced (Fig. 2a-c and Supplementary Fig. 2a). Beyond these expected biomineralization defects, however, the simultaneous deletion of all MFPs also resulted

in the unexpected loss of magnetosome chain formation as magnetosomes are randomly distributed within cells of strain ΔF3 (Fig. 2c). This observation is also reflected by an increased fraction of magnetosomes that have no neighboring particle within a distance of 35 nm and an overall lower number of close-neighboring magnetosomes (Fig. 2d, e and Supplementary Fig. 2b, c).

Individual expression of *mmsF* or *mamF* within ΔF3 restored magnetosome crystal sizes to the level of double deletion mutants whereas complementation with *mmxF* restored crystal sizes incompletely with a certain degree of phenotypic heterogeneity as indicated by the bimodal crystal size distribution (Fig. 2a, c and Supplementary Fig. 2d). In contrast, magnetosome chain formation was reconstituted by expression of *mmsF* and to some extent *mmxF* but not *mamF* indicating that the functions of the MFPs overlap only partially (Fig. 2a, c). Consistently, TEMs of double deletion strains revealed loose magnetosome chains for the Δ*mamF*/*mmxF* and Δ*mamF*/*mmsF* mutants, while Δ*mmsF*/*mmxF*, similar to the *mamF* complemented ΔF3 strain (ΔF3::*mamF*), exhibited aggregates of ferrimagnetic magnetosomes at random cellular positions (Fig. 2a, c and Supplementary Fig. 3a).Thus, only MmsF and MmxF seem to have an additional function during magnetosome chain assembly.

Both, crystal size and magnetosome chain formation are important parameters that influence the cellular magnetic dipole moment and, therefore, strongly affect the cells' ability to align with the geomagnetic field[25]. Consequently, MFP mutants with reduced crystal sizes and defective magnetosome chains are severely impaired in magnetotaxis resulting in homogeneous cell spreading that leads to the formation of round or only slightly elongated swarm halos in semisolid agar in the presence of a magnetic field. In contrast, strains with coherent magnetosome chains and larger crystals show strong magnetic alignments (WT and ΔF3::*mmsF*) resulting in swarm halos strongly elongated in the direction of the magnetic field (Fig. 2f, g).

In summary, these results indicate that MFPs are essential for magnetic navigation and thus organellar function through regulation of magnetosome crystal size and chain formation.

### MFPs mediate MM protein assembly

The absence of magnetosome chains suggests that the expression or function of the magnetoskeletal proteins MamK, MamY, or MamJ might be affected in strain ΔF3. However, all three proteins are present at WT levels in ΔF3 whole-cell extracts and also appear to be functional since fluorescently labelled proteins still localized in WT-like filaments in ΔF3 strains (Fig. 3a, b). We therefore tested whether magnetosome-targeting of the magnetoskeletal proteins is compromised in ΔF3 using Western blots of purified MM extracts. These analyses revealed significantly reduced amounts of MamJ and MamY in ΔF3 magnetosome extracts, while MamK was detectable at levels slightly higher than those of the WT (Fig. 3a). Notably, expression of functional $His_6$-*mmsF* from the strong *mamG* promoter[26] (Supplementary Fig. 3a) mediated MM-targeting of all magnetoskeletal proteins in ΔF3 at elevated levels.

Since MM localization of MamY was previously shown to partially depend on MamJ[16], we surmised that loss of MamJ targeting to the MM is the primary cause for the absence of magnetosome chains in ΔF3. This suggestion is supported by the stronger MamJ-depletion, the phenotypic similarities between Δ*mamJ*, ΔF3::*mamF*, and Δ*mmsF*Δ*mmxF* as well as the absence of MamJ in MMs from strains that lack both *mmsF* and *mmxF* (Fig. 3c and Supplementary Fig. 3a).

Quantitative proteomic analyses of WT and ΔF3 MM extracts revealed eleven proteins that are depleted in ΔF3 magnetosomes. Consistent with the Western blot results, the depleted proteins include MamJ with a 44-fold and MamY with a 6-fold reduced abundance, whereas MamK was identified in equal quantities in WT and ΔF3 MM extracts (Fig. 3d). In addition to MamJ and MamY, the magnetite crystal size-regulating proteins MamD, Mms5, and MamR (84-fold, 12-fold and 9-fold decrease, respectively)[12,27] were also found to be strongly

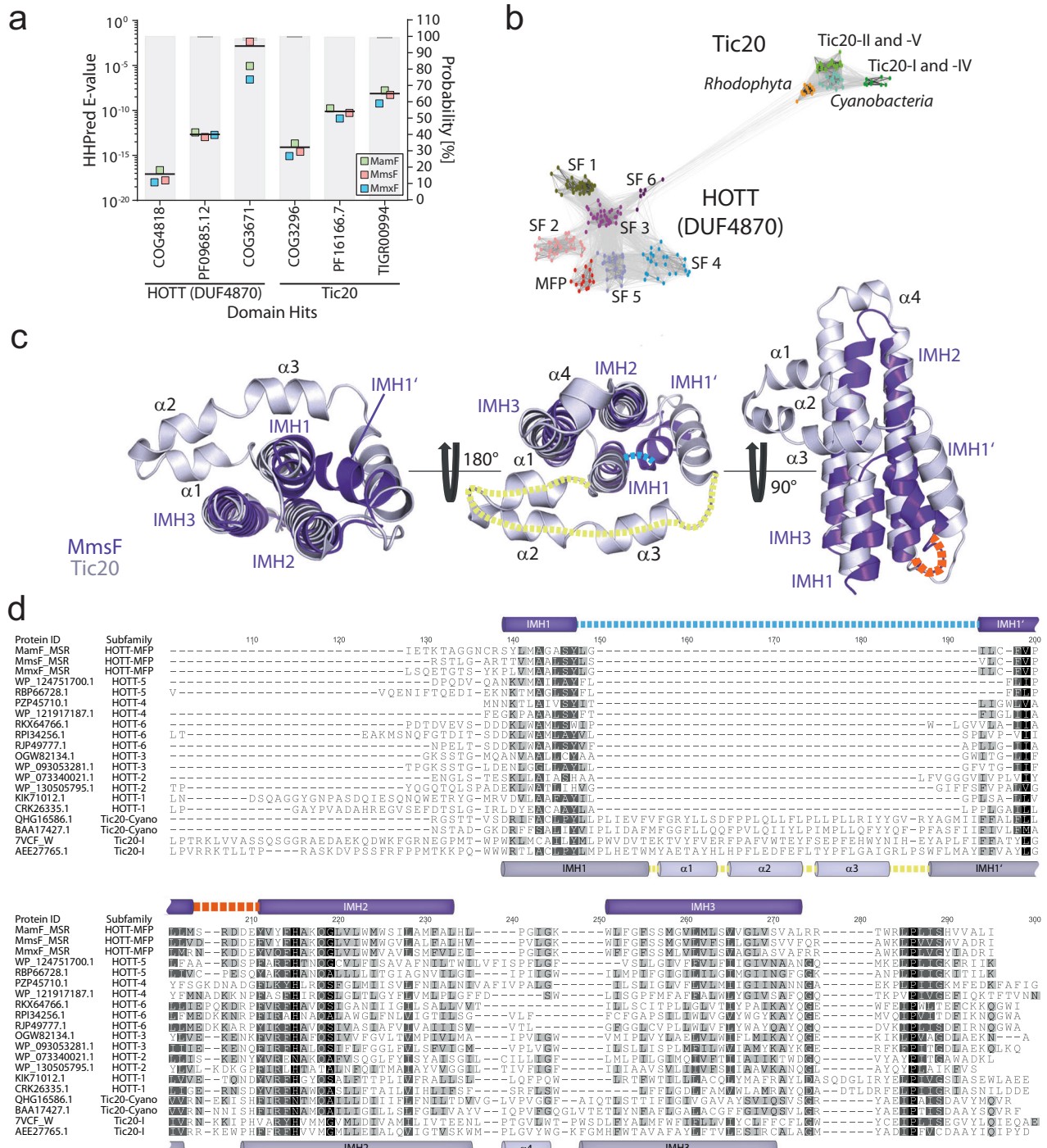

**Fig. 1 | MFPs are members of a common Tic20/HOTT superfamily. a** HHPred analysis of MFPs from MSR-1. E-values (average indicated by a black line) indicate highly significant homologies of MamF, MmsF, and MmxF (green, red, blue squares, respectively) to the HOTT (DUF4870) and Tic20 protein families. The average hit probability of the templates is represented by grey bars with dark grey lines indicating the standard deviation. Only hits with E-values < than $10^{-3}$ are shown. **b** CLANS analysis of 229 Tic20/HOTT superfamily proteins. The HOTT family includes subfamilies (SF) 1-6 and the MFPs. Protein sequences (colored dots) with greater pairwise similarity, are clustered closer together and connected by lines when pairwise BLAST-P E-values are below $10^{-5}$. Accession numbers are listed in Supplementary Data 2. **c** Superimposition of the AlphaFold prediction of the HOTT family member MmsF (MSR-1, AF: A4U547, purple) with the PDB-structure of Tic20 (*Chlamydomonas reinhardtii*, PDB: 7xZI[45], grey) reveals equal architectures (RMSD 2.8 Å, 98 of 107 residues) including a reentrant-like integral membrane helix (IMH1-IMH 1'), followed by a conserved charged loop (dashed orange line) and two

additional integral membrane helices (IMH 2 – IMH 3). All HOTT homologs have a short loop (dashed blue line), while Tic20 representatives contain an extended multi helical loop (dashed yellow line) within the reentrant helix. Superimposition of MamF or MmxF with Tic20 yielded almost identical results (RMSDs are listed in Supplementary Data 2). **d** Multiple aa sequence alignment of Tic20/HOTT subfamily representatives. Amino acid residues are shaded using the similarity color scheme and the Pam250 score matrix. Identical (100%), strongly (80-100%) or weakly (60–80%) similar residues are colored in black, dark grey or light grey, respectively. Positions of predicted or definitive integral membrane helices and alpha helices are indicated by purple (HOTT family) or grey (Tic20 family) cylinders. The reentrant-like helices of MmsF (MSR-1) as well as Tic20 (*C. reinhardtii*) and the conserved acidic amino acids within loop 1 are marked with blue, yellow, and orange dashed lines, respectively. Accession numbers are listed in Supplementary Data 2. **a**, **b**, **d** The source data are provided as a Source Data file.

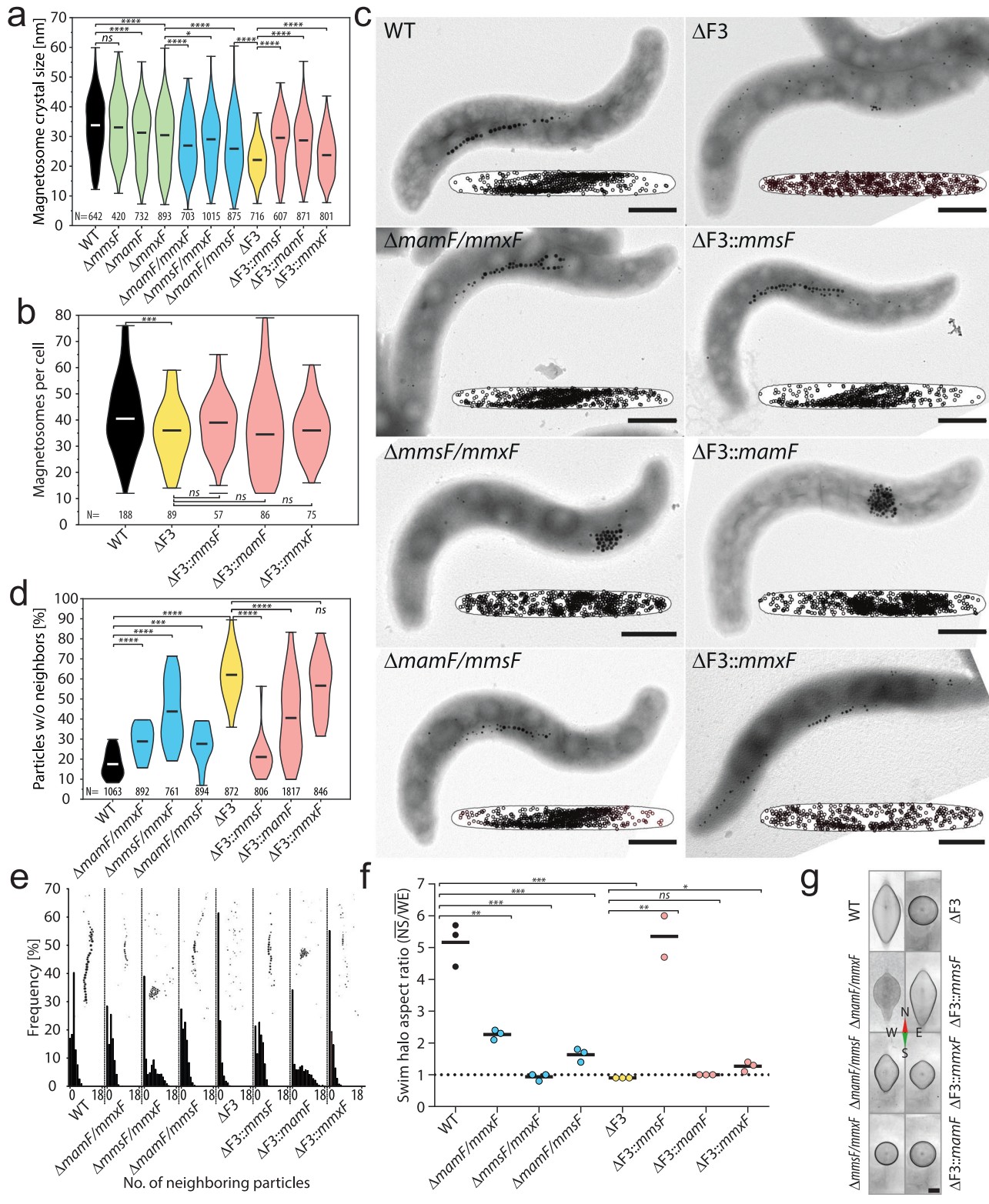

depleted in ΔF3 magnetosomes. In agreement with these results, immunoblotting of isolated MM extracts from strains expressing GFP fusion proteins revealed significantly reduced levels of MamD, Mms5 and MamR in ΔF3 compared to the WT whereas MamG levels were not affected (Fig. 3d and Supplementary Fig. 3b). Interestingly, while MamF fails to mediate MamJ magnetosome targeting (Fig. 3c), in vivo fluorescence imaging of WT and ΔF3 strains producing GFP-tagged MamD, Mms5, or MamR showed that expression of *mmsF*, *mmxF*, or

*mamF* in ΔF3 individually restored magnetosome localization of MamD-, Mms5- and MamR-GFP, respectively (Supplementary Fig. 3c).

These results indicate that all MFPs are central for correct MM assembly but differ in their "substrates".

## MamJ is embedded within the MM

To gain a better understanding of the MFP-mediated MM assembly, we chose to study MamJ as model "substrate" because of its strong

**Fig. 2 | MFPs are essential for magnetosome chain formation and magnetotaxis. a** Deletion of *mamF*-like genes reduces magnetite crystal sizes. Violin plot showing magnetite crystal size distributions of MSR-1 WT (black), single (green), double (blue), and triple (ΔF3, yellow) MFP mutants, as well as complemented ΔF3 strains (red). Sizes measured from TEM micrographs. The number of analyzed particles [N] is indicated below each plot. The min, max, and mean values are indicated by bars. Statistical significance was calculated using a two-tailed Mann-Whitney U test. Asterisks indicate significance: *$P \leq$ 0.05, **$P \leq$ 0.01, ***$P <$ 0.001, ****$P <$ 0.0001, ns, not significant ($P \geq$ 0.05). **b** Deletion of *mamF*-like genes has minimal impact on magnetosome numbers per cell. Violin plot showing magnetosome count distribution per cell. Data and analysis similar to Fig. 2a. **c** Representative TEM micrographs of WT and MFP mutant cells show that magnetosome chain formation is disrupted in the absence of *mmsF* and *mmxF*. The small superparamagnetic magnetite crystals of strain ΔF3 are randomly dispersed throughout the cell. Strains expressing *mamF* only produce larger, ferrimagnetic magnetite crystals which cluster due to magnetic interactions. Insets display normalized intracellular magnetosome distribution of at least 16 cells and 593 particles (scale bars: 500 nm). **d** Quantitative magnetosome neighbor analyses (qMNA; for details see Supplementary Fig. 2b, c) showing the frequencies of magnetosomes without a neighbor within 35 nm. *P*-values were calculated by an unpaired two-tailed *t*-test with Welchs' correction. Data from TEM micrographs. Coloring and statistical significance similar to Fig. 2a. **e** qMNA histograms demonstrating the frequencies of particle neighbor numbers within 35 nm. Insets illustrate representative magnetosome particle distribution schemes. Sample size and coloring similar to Fig. 2d. **f** MFPs are essential for magnetotaxis. Swim halo aspect ratios (N-to-S diameter/W-to-E diameter) within a homogenous 600 μT magnetic field 24 h after inoculation. Dots, colored similar to Fig. 2a, represent aspect ratios of individual biological replicates. Bars represent the mean value. *P*-values were calculated by an unpaired two-tailed *t*-test (labels similar to Fig. 2a). **g** Representative swim halos after 72 hours in a 600 μT magnetic field (scale bar: 1 cm). The data are provided in Supplementary Data 3 and as a Source Data file.

depletion in the ΔF3 mutant and the easily detectable *mamJ* deletion phenotype (Supplementary Fig. 3a). Since MamJ-GFP has a cytosolic localization in the absence of magnetosome proteins (in strain ΔM05, Supplementary Fig. 4a)[28], we initially assumed that MamJ might be membrane-associated by direct interaction with MmsF or MmxF. To assess this assumption, solubilized MM protein complexes were fractionated by size exclusion chromatography (SEC). To reduce complexity, we used MMs of the strain ΔF3Δ*mms5*Δ*mamK*Δ*mamY*::*GFP-mmsF* in which, due to the deletion of all genes encoding MamJ interaction partners like MFPs, MamK, and MamY, merely a complemented, functional GFP-MmsF fusion protein[13] (Supplementary Fig. 4b) can interact with MamJ. In contrast to our expectations, MamJ and GFP-MmsF elution profiles overlapped only partially (Fig. 4a). Thus, only a small fraction of MM-bound MamJ might be directly associated with GFP-MmsF. To evaluate this possibility, solubilized magnetosomes of *GFP-mmsF* and *GFP-mamF* complemented ΔF3 mutant strains as well as the WT and Δ*mamJ* were subjected to co-immunoprecipitation (Co-IP). In these analyses, MamJ was only detected in Co-IP extracts of strain ΔF3::*GFP-mmsF* (Fig. 4b). Nevertheless, even in these extracts only minute amounts of MamJ were detectable as most MamJ remained in the unbound fraction. Thus, only a few molecules of MM-bound MamJ are in direct contact with MmsF. Next, to assess if the majority of magnetosome-bound MamJ molecules might instead be associated with other magnetosome proteins, carbonate extractions with isolated MMs from a Δ*mamK*Δ*mamY* deletion strain were performed. This treatment disrupts protein-protein interactions but leaves protein-lipid interactions intact[29,30]. Therefore, peripherally associated MamJ should be extractable from the MM. Interestingly, unlike the well-known magnetosome-associated protein MamA[31], which could be efficiently extracted from the MM, MamJ, similar to the integral magnetosome protein MamM[32], remained bound to magnetosomes (Fig. 4c). While all tested proteins could be completely extracted from the MM by detergent treatments, even repeated and extended incubations with alkaline, acidic, or high salt buffer solutions failed to solubilize MamJ or integral MamM from the MM (Supplementary Fig. 4c).

Collectively, these results indicate that magnetosome-bound MamJ behaves as an integral membrane protein.

## A hydrophobic C-terminus anchors MamJ to the MM

To investigate which domain of MamJ is required for its MM targeting, alleles encoding full-length (aa 1-426) or truncated *mamJ-GFP*-fusions were expressed in Δ*mamJ*. As shown previously[15], full-length MamJ-GFP restored magnetosome chain formation and localized in a linear pole-to-pole spanning signal. Contrary, GFP fusions to the MamJ N-terminus (aa 1-80) or the alanine-rich central acidic repetitive (CAR, aa 81-333) domain revealed only soluble localization signals (Fig. 4d). When GFP was fused to the MamJ C-terminus (aa 334-426) large fluorescent foci reminiscent of magnetosome clusters could be observed. Since a MamJ334-426-mCherry fusion colocalized with signals from a GFP fusion to the essential magnetosome protein MamB[33] within large fluorescent foci in Δ*mamJ*, the MamJ C-terminus seems to mediate targeting of MamJ to the MM but fails to reconstitute magnetosome chains (Supplementary Fig. 4d). Finally, immunoblotting of isolated magnetosomes from complemented Δ*mamJ* strains confirmed that only full-length MamJ and the MamJ334-426-GFP fusion protein were targeted to the MM (Fig. 4e).

Consistent with its ability to mediate MM targeting, the MamJ C-terminus interacts with MmsF and MmxF, but not MamF, in bacterial two-hybrid (BACTH) assays (Supplementary Fig. 4e) and encompasses a glycine-rich putative integral membrane helix (aa 359-379)[34]. Of note, MamD and Mms5 similarly contain C-terminal integral membrane helices of high glycine content that are preceded by alanine-rich domains. Thus, the three most strongly affected proteins (MamJ, MamD, and Mms5) of the ΔF3 mutant possess a similar domain architecture. While the C-terminal domains of MamD and Mms5 share only low sequence similarity with MamJ, they do also interact with each MFP in BACTH assays (Supplementary Fig. 4f). Thus, in agreement with our fluorescence microscopy results (Supplementary Fig. 3c), all three MFPs facilitate MM targeting of MamD and Mms5. Moreover, the glycine-rich region of MamJ (37.9% glycine content) proved to be essential for magnetosome chain formation as only truncations within or in close proximity of the integral membrane helix (Δ335-358, Δ378-426) prevented reconstitution of magnetosome chain formation in Δ*mamJ*, although all tested variants retained their ability to bind the magnetoskeleton (Fig. 4f). Together with the findings mentioned above (Fig. 4a-e), these results indicate that MamJ is inserted into the MM via its hydrophobic, glycine-rich C-terminus.

## MFPs indirectly facilitate magnetite biomineralization

Previous studies suggested that MFPs directly promote magnetite crystal growth through interaction of iron ions with a cluster of conserved acidic amino acids within a luminal loop (Fig. 1c, d)[14]. Our findings, however, suggest that mistargeting of crystal growth-promoting proteins like MamD or Mms5 is the main contributor for decreased magnetite crystal sizes in MFP mutants. To discriminate between both possibilities, MmsF variants, lacking all acidic (D34N, D36N, D37N, E38N) or charged (D34N, R35Q, D36N, D37N, E38N) residues of the luminal loop were tested for their ability to increase magnetosome crystal sizes in ΔF3 (Fig. 5a). Consistent with an indirect role during magnetite growth, both tested variants restored crystal sizes to the same level as wild-type MmsF (Fig. 5b and Supplementary Data 5). Next, we examined if MmsF maintains its ability to improve crystal growth when most of the proteins mistargeted in ΔF3 are absent. To this end, we deleted *mamR* within the ΔA13Δ*mms5*Δ*mmxF* mutant[35] that already lacks all MFPs and most MFP-dependent proteins

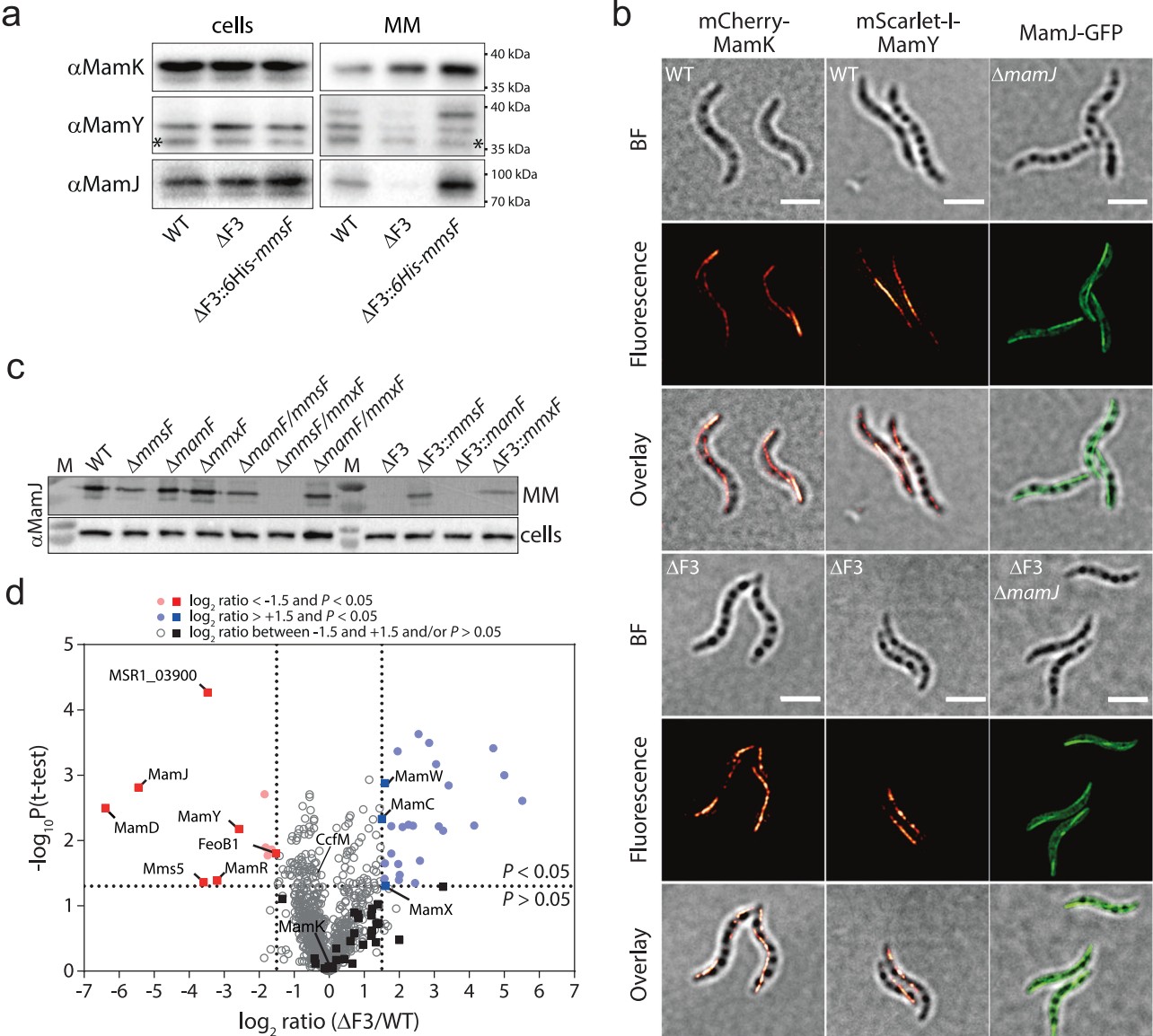

**Fig. 3 | MFPs mediate MM protein assembly. a** Immunodetection of MamK, MamY, and MamJ in whole-cell extracts and isolated MM fractions of the WT, ΔF3, and ΔF3::6His-mmsF strains. Molecular weight standards are indicated. Unspecific bands are marked with asterisks. Loading controls can be found in the source data file. Representative data from two biological replicates are shown.
**b** Magnetoskeletal proteins are still functional in ΔF3. Representative 3D-SIM micrographs showing WT-like filaments of fluorescently labelled MamK, MamY, and MamJ proteins in WT, Δ*mamJ*, and ΔF3 strains. BF, bright field image (Scale bars: 2 µm). Representative data from 3 biological replicates are shown.
**c** Immunodetection of the MM adaptor protein MamJ in whole-cell extracts and isolated MM fractions of the WT, single and double deletion mutants, the triple deletion mutant, and its single MFP complemented strains. MamJ, which is present in all cell extracts, is absent in MM fractions of all strains carrying simultaneous deletions of *mmsF* and *mmxF*. M, molecular weight standard. Electrophoretic

mobility corresponds to ~90 kDa[34]. Representative data from 2 biological replicates are shown. **d** The protein composition of the MM is strongly altered in ΔF3. Volcano plot showing the log₂-fold change of proteins in purified magnetosome fractions of ΔF3 compared to the WT as determined by liquid chromatography–tandem mass spectrometry (LC-MS/MS). MAI-encoded proteins (squares) with significantly altered abundance (log₂ ratio <-1.5 or >1.5 and $P < 0.05$, vertical and horizontal dotted lines, respectively) compared to the WT are labeled. Circles represent non-MAI encoded proteins (e.g. the cytolinker CcfM[55]). Statistical significance was calculated using a two-sided *t*-test. Significantly depleted or enriched proteins are labelled with red and blue symbols, respectively. Non-significant proteins are depicted by grey or black symbols, respectively. Values are the mean value of three biological replicates. The data are provided in Supplementary Data 4 and as a Source Data file.

due to the combined deletion of several accessory magnetosome gene operons (*mamGFDC*, *mms5*, *mms6*, and *mamXY*). Since all essential genes of the *mamAB* operon are maintained, the resulting ΔA13Δ*mms5*Δ*mmxF*Δ*mamR* mutant still formed tiny magnetite particles that are, because of the absence of MFPs, dispersed throughout the cell (Fig. 5c). Upon complementation with *mmsF*, the ΔA13Δ*mms5*Δ*mmxF*Δ*mamR* mutant regained the ability to assemble magnetosome chains but crystal sizes showed only a minor increase

compared to the expression of *mmsF* in the ΔF3 mutant (Fig. 5b, c and Supplementary Data 5). Thus, even though MmsF is functional in the ΔA13Δ*mms5*Δ*mmxF*Δ*mamR* mutant, as evidenced by magnetosome chain restoration, the absence of MFP-targeted proteins prevented considerable magnetite crystal growth.

In summary, our analyses indicate that MFPs indirectly regulate organelle positioning and magnetosome crystal size by mediating correct organelle assembly.

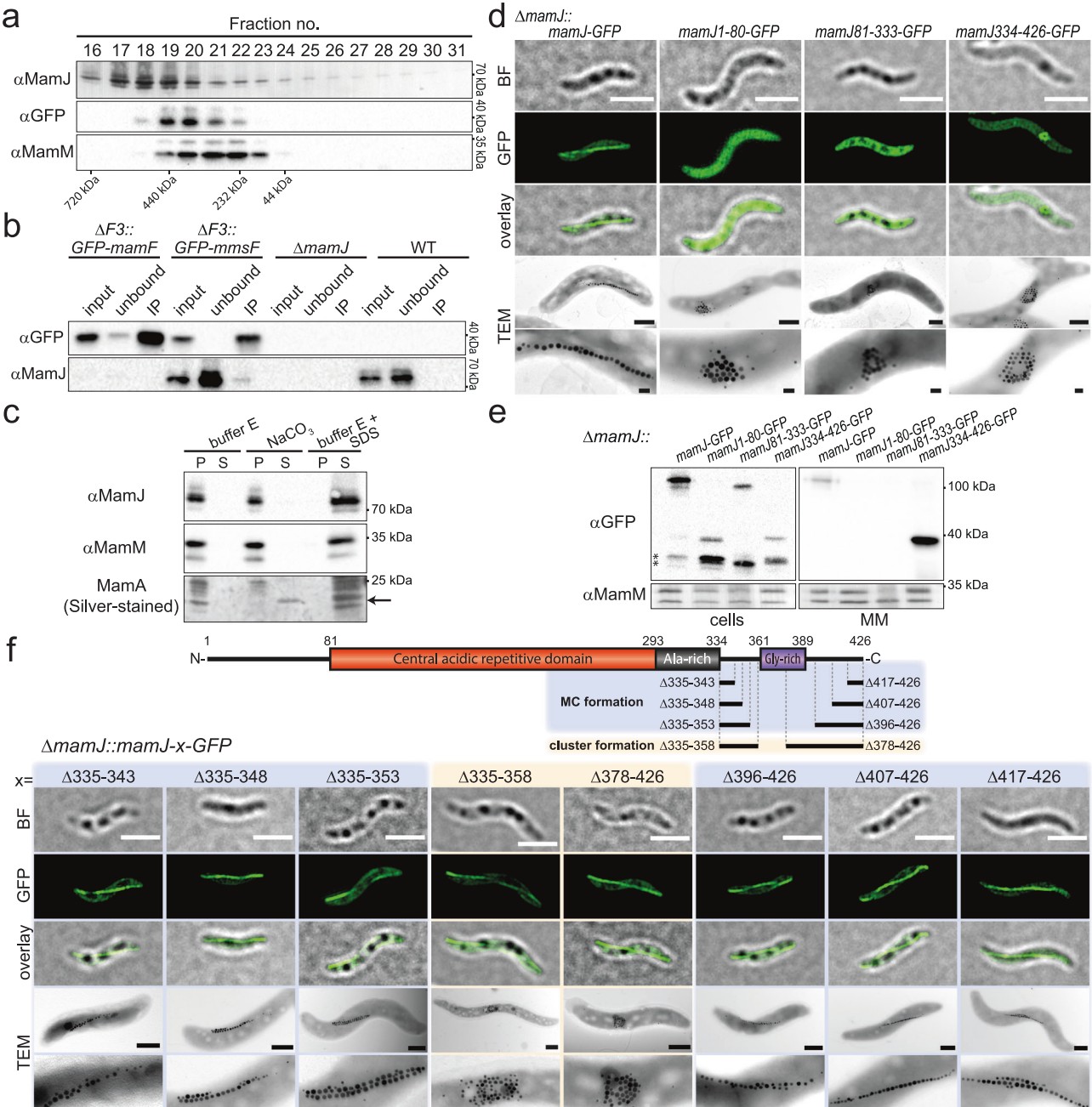

**Fig. 4 | MamJ is anchored to the MM via its C-terminal domain. a** Size exclusion chromatography (SEC) elution profiles of MamJ and GFP-MmsF overlap only partially. Protein complexes from solubilized magnetosomes (ΔF3Δ*mamK*Δ*mamY*::*GFP-mmsF*) were separated using a Superose 6 increase column, and fractions 16-31 were analyzed by SDS-PAGE and Western immunoblots for MamJ, GFP, and MamM. Molecular weights of the SEC standard calibration are indicated. **b** Only minor amounts of magnetosome-bound MamJ directly interact with GFP-MmsF. Co-immunoprecipitation of solubilized magnetosomes from ΔF3::*GFP-mmsF*, ΔF3::*GFP-mamF*, WT, and Δ*mamJ* strains followed by Western immunoblot analysis with GFP and MamJ antibodies. Molecular weight standards are indicated. **c** Magnetosome-bound MamJ behaves as an integral membrane protein. Western immunoblot analysis of buffer E- (negative control), carbonate- (NaCO₃), and SDS-treated (buffer E + SDS, positive control), magnetosome pellet (P) and supernatant (S) fractions from a Δ*mamKY* mutant show that MamJ and the integral MAP MamM are only detectable in supernatant fractions upon treatment with SDS. In contrast, carbonate efficiently extracted the peripheral and highly abundant MamA protein

as shown by silver-stained SDS-PAGE (black arrow). Molecular weight standards are indicated. **d** TEM and 3D-SIM imaging of full-length (aa 1-426) or truncated versions of MamJ fused to GFP and expressed in Δ*mamJ* reveal that only full-length MamJ-GFP is functional to restore the MCs. BF, bright field image (Scale bars: SIM, 2 μm; TEM, 0.5 μm). **e** The C-terminal domain targets MamJ to magnetosomes. Cell extracts and isolated magnetosomes of Δ*mamJ* strains producing GFP-fused MamJ truncations were analyzed by immunoblotting, with MamM as a control. Molecular weight standards are indicated. Asterisks denote unspecific bands. **f** The putative C-terminal integral membrane helix is essential for restoring magnetosome chains. TEM and 3D-SIM imaging of truncated MamJ-GFP variants produced in Δ*mamJ*. Domain structure of MamJ containing a central acidic repetitive (CAR) (red), alanine- and glycine-rich domain (black and purple, respectively). MamJ variants that maintained or lost the ability to reconstitute magnetosome chain formation are colored in blue and apricot, respectively. BF, bright field image (Scale bars: SIM, 2 μm; TEM, 0.5 μm). (**a-f**) Representative data from 2 biological replicates are shown. **a, b, c, e** Source data are provided as a Source Data file.

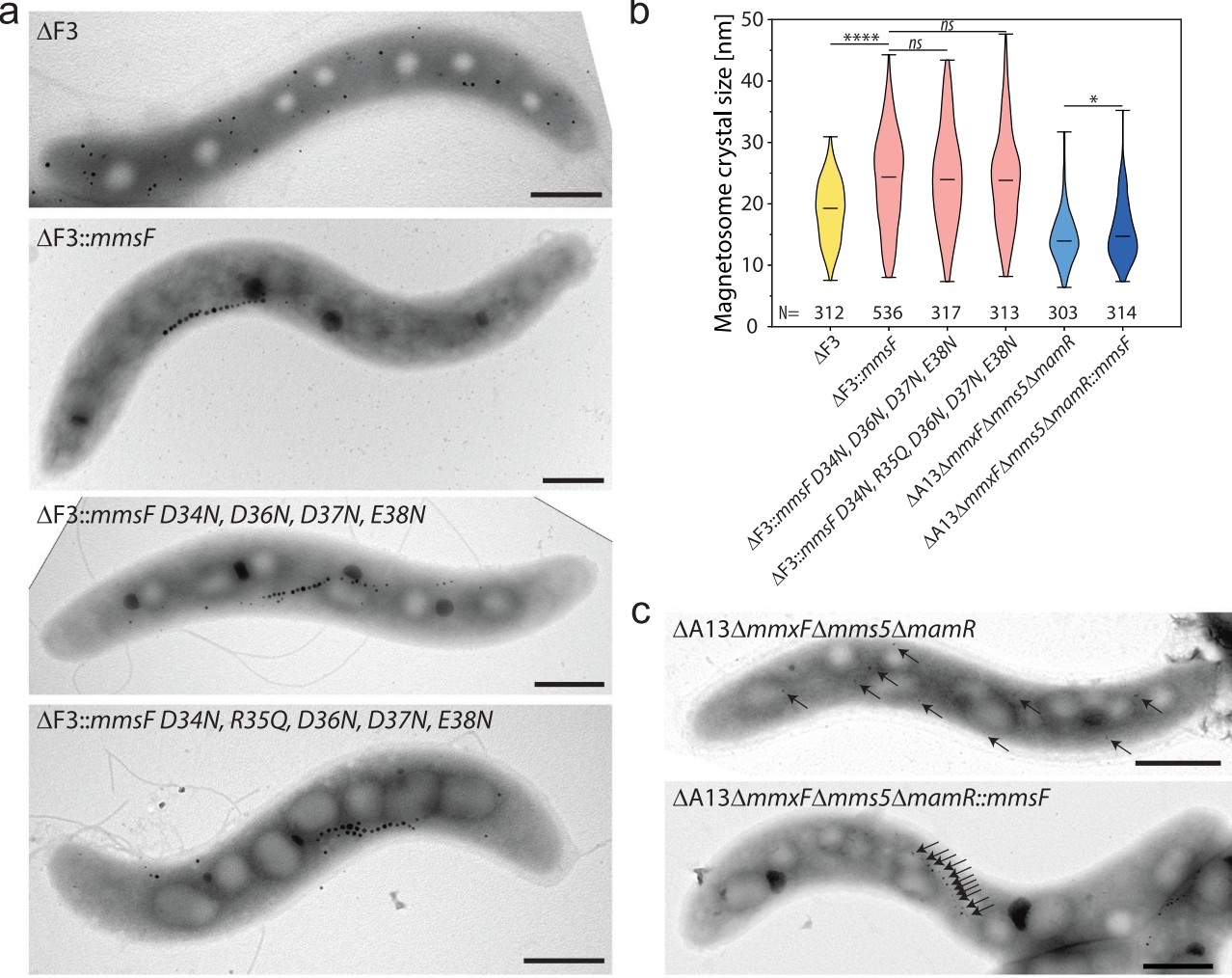

**Fig. 5 | MFPs have an indirect effect on magnetite biomineralization.**
**a** Representative TEM micrographs of ΔF3 and ΔF3 strains expressing *mmsF* wild-type or *mmsF* mutant alleles lacking acidic (D34N, D36N, D37N, E38N) or charged (D34N, R35Q, D36N, D37N, E38N) residues within the loop between integral membrane helices 1 and 2 (see Fig. 1c, d) (Scale bars: 500 nm). **b** Violin plot showing the magnetite crystal size distribution of strain ΔF3 expressing WT and mutant *mmsF* variants as well as strains ΔA13Δ*mms5/mmxF/mamR* and ΔA13Δ*mms5/mmxF/mamR::mmsF*. The number of analyzed particles [N] is indicated. The min, max, and mean values are given by bars. Statistical significance was estimated using an unpaired two-tailed Mann-Whitney U test (*$P \leq 0.05$, **$P \leq 0.01$, ***$P < 0.001$, ****$P < 0.0001$, ns, not significant ($P \geq 0.05$)). Raw data is provided in Supplementary Data 5 and as a Source Data file. **c** Representative TEM micrographs of the ΔA13Δ*mms5/mmxF/mamR* mutant and the complemented strain ΔA13Δ*mms5/mmxF/mamR::mmsF*. The positions of the small magnetosomes are indicated by arrows. The experiments were repeated twice with similar results (Scale bars: 500 nm).

## Discussion

The biogenesis and correct assembly of organelles are crucial for their metabolic functions and, consequently, for overall cellular physiology. While the underlying molecular assembly mechanisms are well characterized in eukaryotes they remained poorly understood in prokaryotes, even though their organelles are typically less complex. Here, we discovered that MamF-like proteins play essential roles in the assembly and function of magnetosome organelles in magnetotactic bacteria. Upon deletion of all *mamF*-like genes in MSR-1, several magnetosome proteins (e.g. MamJ, MamD, Mms5) are mislocalized which causes magnetosome mispositioning, reduced magnetite biomineralization as well as loss of magnetotaxis (Fig. 2). These findings not only mark an essential step towards a mechanistic understanding of magnetosome biogenesis, but also redefine the roles of MFPs. While previous studies assumed an active magnetite crystal growth-promoting activity[13,14], our data show that MFPs fail to significantly improve magnetite crystal growth in the absence of their "substrates" (Fig. 5). The contribution of the MFPs to magnetite crystal maturation thus represents only an indirect effect that is based on the direct MM targeting of proteins which themselves mediate crystal growth (MamD and Mms5).

The detailed analysis of the potential substrate protein MamJ, revealed a cytosolic localization in the absence of MFPs whereas it is tightly bound to the MM in their presence, necessitating the use of detergents for extraction (Figs. 3c and 4c and Supplementary Fig. 4a, c). Notably, MamJ membrane-binding appears to be largely independent of protein-protein interactions as magnetosome-bound MamJ is resistant against carbonate, high salt, acidic, or alkaline treatments (Fig. 3a, b and Fig. 4a-c and Supplementary Fig. 4c). Since similar results were obtained for the integral MM protein MamM, our results suggest that MamJ may also be membrane integral. While MamJ membrane integration could potentially depend on protein lipidation, no evidence for this posttranslational modification was obtained by our proteomic analysis. Instead, we could show that magnetosome targeting of MamJ depends on a putative C-terminal integral membrane helix (aa 359-379)[34] which represents one of the very few conserved regions within this hypervariable protein[36]. Only truncations that partially deleted the putative IMH (Δ378-426) or potentially interfered with its accessibility or folding (Δ335-358) disrupted magnetosome chain reconstitution by MamJ, likely by preventing its MM binding (Fig. 4d-f and Supplementary Fig. 4d, e). Our results thus

suggest that MFPs mediate membrane insertion of otherwise soluble proteins.

MamJ is crucial for magnetosome chain formation by tethering magnetosomes to the cytosolic MamK filament (Supplementary Fig. 3a)[15]. Given the almost complete absence of MamJ from the magnetosome membrane of strain ΔF3, we anticipated a significant reduction in MamK binding to magnetosomes. Surprisingly, our proteomic and immunoblot analyses revealed wildtype-like MamK levels on the magnetosome membrane of ΔF3. This finding suggests that MamK interacts with additional magnetosome proteins beyond MamJ. However, no such interactions have been reported to date. We thus hypothesize that a protein with unaffected or slightly elevated abundance in the ΔF3 magnetosomes may also have the capacity to bind MamK, albeit weakly, as evidenced by the lack of chain formation in this strain. Further investigations will be necessary to identify such potential MamK-binding partners and clarify their roles in magnetosome organization.

Interestingly, MamD and Mms5, two further potential MFP substrates, share several characteristics with MamJ. Both proteins can also only be extracted from magnetosomes by use of detergents[37] and possess a soluble, alanine-rich N-terminal domain followed by a moderately hydrophobic (grand average hydropathy scores 1.3-1.6[38],), glycine-rich C-terminal domain (Supplementary Fig. 4f). While the Ala-rich domains lack any sequence homology, the C-terminal domains share an integral membrane helix with a GxxxG glycine zipper motif that is commonly found within membrane proteins[39]. However, the GxxxG motif is unlikely to serve as a universal recognition motif for MFPs as the partial MamD paralogs MamG and Mms6 harbor similar motifs but do not interact with MFPs and remain largely unaffected by their deletion (Supplementary Fig. 3b, c and Supplementary Fig. 4g). Interestingly, recent reports have shown that the magnetosome localization of MamD (and Mms6) is dependent on biomineralization-permissive conditions (i.e. microoxic conditions) in the closely related magnetotactic bacterium *Magnetospirillum magneticum*[10,40]. This biomineralization-dependent MM sorting relied on an intact MM growth checkpoint, consisting of the protease MamE, its activators MamO and MamM, as well as the negative regulator MamN[10]. Our findings therefore indicate that MFP-mediated magnetosome targeting of MamD might occur after its proteolytic processing by MamE and release from an inhibition by MamN[10]. However, further studies are required to determine how MFPs exactly recognize their substrates and how their targeting is regulated. In the absence of major structural changes between the MFPs it would also be of special interest to identify the molecular basis of their different "substrate" specificities. In the future, this knowledge could be exploited to sustainably produce tailored, multifunctional magnetosomes for use in biotechnology and biomedicine (e.g. as contrast agents in medical imaging, for magnetic hyperthermia treatments, or magnetic drug targeting)[41,42].

Here, we further demonstrated that the ability of magnetobacterial cells to align with an external magnetic field mainly depends on their capacity to form magnetosome chains. Swarm halos elongated in the direction of the magnetic field were exclusively observed in strains capable of at least rudimentary magnetosome chain formation (e.g. ΔF3::*mmxF*). In contrast, magnetosome chain-free strains were unable to align in the magnetic field and formed only equidimensional swarm halos (ΔF3, ΔF3::*mamF*, and Δ*mmsF/mmxF*). Notably, we also observed an unexpected difference between the isogeneic strains ΔF3::*mmsF* and Δ*mamF/mmxF*. Although both strains express *mmsF* as the sole MFP, only ΔF3::*mmsF* showed a wildtype-like magnetotactic behavior. We attribute this difference to the stronger expression of *mmsF* in strain ΔF3::*mmsF* due to the use of a stronger promotor ($P_{mamG}$ vs. the natural $P_{mms6}$ in strain Δ*mamF/mmxF*)[26]. Elevated *mmsF* expression in ΔF3::*mmsF* likely improves the targeting of biomineralization proteins such as MamD and Mms5, resulting in larger, more magnetic crystals and enhanced cohesion within the magnetosome chains (Fig. 2a, c, e). This increased cohesion, accompanied by a reduced number of "free" magnetosomes, is likely due to enhanced recruitment of MamJ (Fig. 3a). The swarm agar assay may thus be sensitive enough to quantify the relative contributions of magnetosome chain formation and magnetite crystal size for magnetotaxis. Such experiments, however, would require strictly controlled conditions due to the growth-dependent establishment of nutrient and oxygen gradients that caused slight differences in formation of elongated swarm halos during our experiments (Fig. 2f, g).

Using various bioinformatic tools, we showed that MFPs of magnetotactic bacteria are members of the HOTT (DUF4870) protein family, which itself constitutes a superfamily with the Tic20 protein family (Fig. 1 and Supplementary Fig. 1). While the HOTT family is completely uncharacterized yet, eukaryotic representatives of the Tic20 family are known to mediate the import of proteins into plastid organelles[19–21,43]. In plants, Tic20 has been proposed to be the major preprotein-translocating channel subunit of the Tic complex located at the inner envelope of chloroplasts[22,43]. Based on this supposed Tic20 function and the ability of MFPs to assemble into larger oligomers[14], it is tempting to speculate that MFPs might form channels that facilitate the assembly of magnetosomes by inserting proteins into the MM that are not recognized by generic membrane protein integrases (e.g. the Sec translocon). Such a function could explain why MamJ or MamD have a cytosolic localization in the absence of MFPs but require detergents for their extraction from the MM when MFPs are present. The observed physical interaction between MFPs and the IMH-containing C-terminal domains of their substrates may thus only be required during the initial steps of MM insertion which would also explain why only a minor fraction of MamJ is found in complex with MmsF (Fig. 4a,b and Supplementary Fig. 4e). However, further studies are needed to clarify the molecular mechanism of MFP-mediated magnetosome assembly, particularly given recent findings from two plastidic TOC-TIC super-complex structures, which revealed that Tic20 is only one component of a larger heteromeric translocation channel, leaving its precise role still in question[44,45]. Since MFPs and Tic20 share a similar structural architecture, and both play essential roles in proper organelle assembly, future mechanistic analyses of the MFP-mediated magnetosome protein targeting may also help to elucidate the role of Tic20 during chloroplast development.

Initially, we were surprised to identify homologs of the plastidal biogenesis factor Tic20 within an alphaproteobacterium that is distantly related to the ancestor of mitochondria. However, in contrast to Tic20, whose occurrence is restricted to plastid-containing eukaryotic linages (11 phyla) and cyanobacteria, the homologous HOTT family is widely spread among all domains of life (present in at least 38 bacterial, 13 archaeal, and 5 eukaryotic phyla). This extensive phylogenetic distribution and the larger number of divergent subfamilies suggest that the HOTT family is evolutionarily more ancient. Thus, although short protein lengths, long evolutionary distances, and high sequence divergences among superfamily representatives[21,46] prevented more precise phylogenetic reconstructions (Supplementary Fig. 1c, d), our findings support the hypothesis that Tic20 originated from a HOTT ancestor that likely already mediated protein targeting. In agreement with this suggestion, conserved domain database analyses revealed simultaneous hits to Tic20 and the HOTT domain within subfamily 2 that contains all cyanobacterial HOTT representatives (Supplementary Fig. 1f). In contrast, previous studies suggested that Tic20, similar to the mitochondrial inner membrane preprotein translocases (Tim17/22/23), evolved from bacterial LivH-like amino acid transporters[47–49]. In support of our data, however, several subsequent studies failed to detect any significant homologies between LivH, Tic20, or Tim17/22/23, respectively[50–52].

In summary, we not only present the functional characterization of members of the yet uncharacterized HOTT protein family but also provide insights into its evolutionary relationship with the Tic20

protein family. Therefore, our findings suggest the presence of at least primitive organelle-specific protein integrases or translocases within bacteria and shed new light on the evolution of eukaryotic organelles.

## Methods

### Media and buffers
All media and buffers used in this study are listed in Supplementary Table 4.

### Bacterial strains and culture conditions
All strains used in this study are listed in Supplementary Table 1. Strains of *Escherichia coli* were grown in lysogeny broth (LB) supplemented with kanamycin ($25\,\mu g\,mL^{-1}$) or ampicillin ($50\,\mu g\,mL^{-1}$) at 37 °C (except BTH101 which was grown at 28 °C) and shaking at 170 rpm. For cultivation of *E. coli* WM3064, 0.1 mM DL-α,ε-diamino-pimelic acid (DAP) was added. Unless stated otherwise, MSR-1 strains were grown microaerobically at 28 °C in modified flask standard medium (FSM, pH 7, 10 mM HEPES, 12 mM potassium lactate, 4 mM $NaNO_3$, 0.74 mM $KH_2PO_4$, 0.60 mM $MgSO_4 \cdot 7\,H_2O$, $50\,\mu M$ Fe(III)-citrate, 0.01% (w/v) yeast extract, 0.3% (w/v) soybean peptone) with moderate agitation (120 rpm). For selection of chromosomal insertions, tetracycline ($20\,\mu g\,mL^{-1}$), chloramphenicol ($5\,\mu g\,mL^{-1}$) or kanamycin ($5\,\mu g\,mL^{-1}$) were added.

### Molecular and genetic techniques
Molecular techniques were performed using standard protocols[53]. All constructs were sequenced by Macrogen Europe (Amsterdam, Netherlands).

### Plasmids and primers
All plasmids and primers used in this study are listed in Supplementary Table 2 and Supplementary Table 3, respectively.

### Generation of site-specific chromosomal insertion and deletion mutants
For the generation of unmarked deletion mutants, ~0.7 kb fragments of the up- and downstream flanking regions of the corresponding gene were amplified by PCR using Phusion polymerase (NEB) and primers. After gel purification of PCR products, the corresponding up- and downstream fragments were fused by overlapping extension PCR using T4 polynucleotide kinase (Thermo Scientific) phosphorylated primers. Fused PCR products were then cloned into an EcoRV-linearized and dephosphorylated pORFM-GalK-MCS deletion vector. Plasmids were subsequently transferred to MSR-1 by conjugation using *E. coli* WM3064 as a donor[32]. Therefore, cultures of plasmid-containing *E. coli* and receptor MSR-1 strains were cultivated in LB or FSM media over-night before $2\times10^{-9}$ cells of each strain were mixed in 15 ml falcon tubes, pelleted at 5,421 xg for 10 minutes (Allegra X-15R centrifuge; SX4750A rotor, Beckman Coulter, Brea, USA) and spotted onto FSM agar plates. After over-night incubation at 28 °C and 2% $O_2$, cells were resuspended in 10 ml of FSM, grown in 15 ml falcon tubes at 28 °C for 2 hours at 120 rpm agitation, pelleted again and plated on selective FSM agar plates. After incubation for five days (28 °C, 2% $O_2$), kanamycin-resistant plasmid insertion mutants were transferred to $100\,\mu L$ fresh FSM, and grown overnight (28 °C, 2% $O_2$). $100\,\mu l$ cultures were then inoculated into $900\,\mu L$ fresh FSM and incubated for 24 h at 28 °C and 2% $O_2$ before $200\,\mu L$ were plated on FSM agar plates containing 2.5% (w/v) galactose and $100\,ng\,mL^{-1}$ anhydrotetracycline. After incubation for additional 5 days (28 °C, 2% $O_2$), mutant colonies were transferred to $100\,\mu L$ FSM and verified by PCR.

### Construction of MSR-1 expression vectors
For complementation of *mamF*-like mutant strains, *mamF*, *mmsF*, and *mmxF* were PCR-amplified with specific primers, digested with KpnI/SacI, and cloned into similarly digested pBAM-Tet-*mamD*-GFP vector[54] to replace *mamD-GFP* against *mamF*-like genes. The resulting constructs were then transferred to MSR-1 mutants via conjugation. For expression of *mamF*-like-*GFP* fusions, *GFP* together with a helix linker was amplified by PCR from pBAM160-*GFP-ccfM*[55], digested with KpnI and cloned into KpnI-digested pBAM-Tet-*mmsF* and pBAM-Tet-*mamF* vectors, respectively. The resulting plasmids pBAM-Tet-*GFP-mmsF* and pBAM-Tet-*GFP-mamF* were then transferred to MSR-1 strain via conjugation.

For expression of additional fluorescently labeled proteins in MSR-1, *mamR*, *mms5*, and *mamG* were PCR-amplified with specific primers (Supplementary Table 3), digested with KpnI/XbaI, and cloned into the similarly digested pBAM-Tet-*mamD-GFP* vector[54] to replace *mamD*. For co-expression with *mamF*-like genes, *mamD-GFP*, *mamG-GFP*, *mamR-GFP*, and *mms5-GFP* containing pBAM-Tet vectors were digested with EcoRI/SacI to cut out the complete expression cassette including the constitutive *mamG* promotor. The resulting DNA fragments were then cloned into EcoRI/SacI-digested pBAM1[56] vector containing a kanamycin resistance gene. These constructs were then transferred to MSR-1 mutants via conjugation. For co-expression, transfers were performed successively in independent conjugation experiments.

For expression of *mCherry-mamK*, the *mCherry-mamK* construct was amplified from MSR-1::*mCherry-mamK* containing a chromosomal in frame fusion[57]. The PCR product was digested with NdeI/XbaI and ligated into NdeI/XbaI-digested pBAM-$P_{tet}$-*mamA-GFP* for exchange against *mamA-GFP*[55]. Similarly, a synthetic *mScarlet-I-mamY* (ATG:biosynthetics GmbH, Merzhausen, Germany) was inserted into the NdeI/XbaI restrictions sites of pBAM-$P_{tet}$-*mamA*-GFP to yield pBAM-$P_{tet}$-mScarlet-I-*mamY*.

For the expression of truncated *mamJ-GFP* fusions, selected regions were PCR-amplified and cloned into the NdeI/KpnI restriction sites of the vector pBAM-Tet-*mamJ-GFP*[58]. Alternatively, an inverse PCR-based cloning strategy was used to generate C-terminal truncations.

To study the colocalization of the MamJ C-terminus with MamB-GFP, a *mamJ*334-426-*mCherry* fusion was generated. Therefore, the vector pBAM2-Tn7-Tet-$P_{lac}$[59] was digested with NotI/XhoI. The $P_{lac}$-*mamB_lacI* containing fragment was then ligated into the NotI/XhoI sites of pBAM2-Tn7-Cam (Uebe, unpublished). Subsequently, a PCR-generated *mamJ*334-426-mCherry fusion was inserted into the NdeI/NotI restrictions sites to exchange *mamB* and yield pBAM2-Tn7-Cam-$P_{lac}$-*mamJ*334-426-mCherry.

All constructs were verified by PCR and DNA sequencing.

### Motility soft agar assay
Soft agar (0.2% agar (w/v)) swimming assays were conducted, using modified FSM with a reduced concentration of potassium lactate (Supplementary Table 4)[25]. Precultures were grown in triplicates per strain for at least three passages prior to the experiment in 6 well plates in FSM at 28 °C under defined microoxic condition (2% $O_2$). Five microliters of cell suspension (adjusted to an $OD_{565}$ of 0.1) were pipetted into 7 mL soft agar within 6-well plates and incubated at 28 °C under ambient oxygen conditions within a homogenous magnetic field of 0.6 mT using a custom coil setup. The plates were documented after one and two days of incubation and the swimming expansion was measured. For better visualization of the swim halos, the experiment was repeated in 90 mm petri dishes under otherwise identical conditions and documented after 72 h.

### Bacterial two-hybrid assays
For protein interaction studies using the adenylate cyclase-based bacterial two-hybrid assay[60], genes of interest were amplified from isolated chromosomal DNA of MSR-1 and cloned into pUT18C, pUT18, pKT25, and pKNT25 plasmids. Therefore, the respective primers were phosphorylated before amplification using T4 polynucleotide kinase (Thermo scientific). Phosphorylated PCR-fragments were then ligated with SmaI-linearized and dephosphorylated (FastAP, Thermo

scientific) vectors. The resulting T18- and T25-based plasmids were co-transformed into the *E. coli* BTH101 reporter strain. Cells were plated on LB agar supplemented with 40 µg mL⁻¹ 5-bromo-4-chloro-3-indolyl-β-D-galactopyranoside (X-Gal), 0.5 mM isopropyl β-D-1-thiogalactopyranoside (IPTG), ampicillin (100 µg mL⁻¹) and kanamycin (50 µg mL⁻¹) and incubated at 28 °C for 24 h. Finally, several colonies per plasmid combination were grown overnight at 28 °C in LB medium supplemented with IPTG (0.5 mM), ampicillin (100 µg mL⁻¹) and kana-mycin (50 µg mL⁻¹), and 3 µL of culture were spotted onto M63 mineral salts agar containing 0.2% (w/v) maltose, X-Gal (40 µg mL⁻¹), 0.5 mM IPTG, ampicillin (50 µg mL⁻¹), and kanamycin (25 µg mL⁻¹) (Supplementary Table 4)[61]. M63 plates were then incubated at 28 °C for one day and documented with a Lumix FZ38 camera (Panasonic, Kadoma, Japan). Solely combinations with homogeneous, intense blue color formation were regarded as positive. Co-transformants harboring empty vectors or combinations of empty and test vectors of the respective T18- and T25-protein fusions served as negative controls on the same plates. All experiments were carried out in triplicates.

## Magnetosome isolation

For magnetosome isolation, MSR-1 strains were cultivated for 24 h at 28 °C and 120 rpm in 5 or 10 L flasks filled with 3 or 5 L FSM, respectively. For comparison of the MM proteome, MSR-1 strains were cultivated by anaerobic fed-batch fermentation in 3 L Eppendorf bioreactors using a BioFlo 320 system (Eppendorf, Hamburg, Germany)[62,63]. To regulate the pH, 1 M KOH or an acidic feed composed of 1 M HNO₃, 0.85 M potassium lactate, 0.45 M sodium nitrate, 25 mM magnesium sulfate and 3 mM Fe(III)-nitrate was pumped into the bioreactor when the pH changed by 0.1 from the set point of pH 7.0. After 30-35 h, cells were harvested by centrifugation at 8000 xg at 4 °C for 15 minutes (Avanti J−26 XP centrifuge; JLA 8.1000 rotor, Beckman Coulter, Brea, USA). Pellets were washed with buffer W (20 mM 4-(2-hydroxyethyl)-1-piperazineethanesulfonic acid (HEPES), 5 mM ethylenediaminetetraacetic acid (EDTA), pH 7.4) twice, pelleted by centrifugation, and stored at -20 °C until further use.

Magnetosome isolation was performed with an optimized protocol described recently[41]. Briefly, cell pellets were resuspended in ~35 mL g⁻¹ in buffer R (50 mM HEPES, 1 mM EDTA, 0.1 mM phenylmethylsulfonyl fluoride (PMSF), pH 7.4; Supplementary Table 4) and lysed by three cycles in a M110-L Microfluidizer processor (Microfluidics Corp., Westwood, USA) equipped with a H10Z interaction chamber at 1241 bar. Lysed cells were cleared from cell debris by centrifugation at 750 xg at 4 °C for 10 minutes (Allegra X-15R centrifuge; SX4750A rotor, Beckman Coulter, Brea, USA). The supernatant was loaded onto a magnetized MACS CS column (Miltenyi Biotec, Bergisch Gladbach, Germany) equilibrated in 50 mL of buffer E (10 mM HEPES, 1 mM EDTA, pH 7.4). The column was washed with 50 mL of buffer E, followed by 50 mL of buffer S (10 mM HEPES, 1 mM EDTA, 200 mM NaCl, pH 7.4) and again 50 mL of buffer E. Afterwards, the magnets were removed from the column and the magnetic fraction was eluted with 30 mL of ddH₂O. Afterwards, HEPES and EDTA were added to a concentration of 10 mM and 1 mM, respectively, and 15 mL of this magnetic fraction was centrifuged on top of a 7 mL 60% (w/w) sucrose cushion in buffer E at 4 °C, 183,700 xg for 1.5 hours (Sorvall WX Ultra 80, Thermo Fisher Scientific, Waltham, MA, USA)[64]. The supernatant was discarded and the magnetosome pellet resuspended in 200 µL buffer E.

## Protein concentration determination and normalization of magnetosome solutions

Protein concentrations were determined with the Roti-Quant universal kit according to the manufacturer's protocol in flat-bottomed 96-well plates using an Infinite M200Pro plate reader (Tecan Group Ltd., Männedorf, Switzerland).

## Gel electrophoresis and Western immunoblot

Analysis of proteins by denaturing sodium dodecyl sulfate polyacrylamide gel electrophoresis (SDS-PAGE) was performed on 8×10 cm or 10.5×10 cm acrylamide gels with 1.5 mm thickness at 300 V and 25 mA per gel (Hoefer SE250/SE260 chamber, Pharmacia Biotech, Upsala, Sweden; MP-300V power supply, Major Science, Saratoga, USA). If not stated otherwise, samples corresponding to 10 µg protein were mixed with 5x SDS-sample buffer (10% (w/v) SDS, 25% (v/v) β-mercaptoethanol, 25% (v/v) glycerol, 0.05% (w/v) bromophenol blue, 0.3 M Tris/HCl pH 6.8) to a 1-fold concentration and incubated at 95 °C for 10 minutes (ThermoMixer F1.5, Eppendorf, Hamburg, Germany) prior loading to the gels. For standard SDS-PAGE, 8-22.5% or 12-22.5% acrylamide linear gradient gels were used. To visualize protein bands, gels were stained with Coomassie brilliant blue R-250. To this end, gels were incubated for 30 minutes in Coomassie staining solution (50% (v/v) methanol, 10% (v/v) acetic acid, 0.1% (w/v) Coomassie R-250) followed by incubation in washing solution (10% (v/v) methanol, 7% (v/v) acetic acid) over-night. Alternatively, an improved silver staining method was used[65]. To this end, gels were incubated in fixing solution (40% (v/v) methanol, 36.5% (v/v) of 37% (w/v) formaldehyde) for 10 minutes, washed twice in ddH₂O for 5 minutes, incubated with 0.02% (w/v) thiosulfate solution for 1 minute before washing with ddH₂O for 20 s twice. The gel was then covered with impregnating solution (0.1% (w/v) AgNO₃) for 10 minutes in the dark. After shortly washing in ddH₂O, a small volume of developing solution (3% (w/v) Na₂CO₃, 0.135% (v/v) of 37% (w/v) formaldehyde, 0.02% (v/v) of 0.02% (w/v) Na₂S₂O₃) was used for washing the gel and then removed. The gel was incubated with fresh developing solution until protein bands became visible (1 to 5 minutes). The development was stopped by discarding the developing solution and incubating the gel in stopping solution (1.86% (w/v) EDTA) for 10 minutes. Stained gels were documented in ChemiDoc XRS+ (Bio Rad Laboratories Inc., Herkules, USA) or a DMC-FZ38 digital camera (Panasonic Corp., Kadoma, Japan).

Western blotting and immunodetection were performed according to protocols described previously[28].

## Protein complex purification

For analysis of magnetosome protein complexes by size exclusion chromatography, isolated magnetosomes of MSR-1 ΔF3Δmms5ΔmamKΔmamY::GFP-mmsF were solubilized using 0.1% (w/v) lauryl maltose neopentyl glycol (LMNG). To this end, isolated magnetosomes were adjusted to an OD₄₉₂ of 6 in buffer E to a final volume of 10 mL and pelleted by centrifugation at 21,130 xg and 4 °C for 30 minutes. The pellet was resuspended in 2.5 mL of buffer E containing 50 mM NaCl, 0.1% (w/v) LMNG and incubated over-night at 4 °C under constant agitation. Subsequently, magnetosomes were removed from this solution by centrifugation at 21,130 xg and 4 °C for 30 minutes. The supernatant containing the solubilized protein complexes was then subjected to an ultracentrifugation on top of a 98% (w/v) glycerol cushion at 385,900 x g, 4 °C for 30 minutes (Optima MAX XP centrifuge, Beckman Coulter, Brea, USA; MLA130 rotor, Beckman Coulter, Brea, USA). The resulting supernatant was carefully removed and centrifuged once more at the same ultracentrifugation settings without a glycerol cushion. The resulting, magnetosome-free supernatant was then concentrated using Amicon Ultra 0.5 mL spin columns with a 3 kDa molecular weight cut-off at 4 °C according to the manufacturer's protocol to a protein concentration of 3.5 mg mL⁻¹ in a volume of 300 µL. 100 µL of this solution were then separated on a Superose 6 Increase 10/300 GL column with a flow rate of 0.3 mL per minute on an Äkta pure system (GE Healthcare, Uppsala, Sweden). Chromatography was performed in buffer E containing 50 mM NaCl, 0.005% (w/v) LMNG and 0.5 mL fraction were collected. 15 µL were then used for analysis by SDS-PAGE and Western blotting.

## Extraction of magnetosome proteins

Carbonate extractions were performed according to a modified protocol described previously[66]. For carbonate extraction of MM-associated proteins, 400 μL of magnetosomes with an $OD_{492}$ of 6 in buffer E (corresponding to 0.28 μg mL$^{-1}$ protein) were pelleted by centrifugation (20 min, 21,000 xg, 10 °C). The supernatant was carefully discarded, and the pellet was resuspended in 400 μL 100 mM NaCO$_3$ (pH 11.3) by vortexing. Samples containing 400 μL buffer E or 400 μL of 1% SDS in buffer E, served as negative and positive controls, respectively. After incubation for 1.5 h at RT on a roll mixer, samples were pelleted by centrifugation (20 min, 21,000 xg, 10 °C). The supernatants were transferred to ultracentrifuge tubes, refilled with respective buffers (ad 600 μL) and centrifuged (30 minutes, 385,900 xg, 4 °C; Optima MAX XP centrifuge, Beckman Coulter, Brea, USA; MLA130 rotor, Beckman Coulter, Brea, USA) twice. While the magnetosome pellets were resuspended in the same buffers again, the supernatant was neutralized and precipitated in 150 μL aliquots by addition of 1.5 mL ice-cold 90% (v/v) acetone, 10% (v/v) trichloroacetic acid (TCA), 10 mM NaCl, and incubation at -20 °C overnight. Supernatant samples were then centrifuged at 21,000 xg 4 °C for 20 min and the supernatant was discarded. Precipitates were washed with 1 mL ice-cold acetone and pelleted by centrifugation (21,000 xg, 4 °C for 20 min) three times. Pellets were then dried at RT to remove residual acetone, resuspended in 40 μL buffer E and 5x SDS loading buffer with short incubation in a sonication bath, and stored at 20 °C until further use. Resuspended magnetosomes were washed in buffer E three times before magnetosome pellets were resuspended in 300 μL buffer E and 5x SDS loading buffer and stored at 20 °C until use. 20 μL aliquots were used for SDS-PAGE and subsequent Western blot analyses.

Further extractions were performed with buffer E and basic (0.1 M N-cyclohexyl-3-aminopropanesulfonic acid (CAPS-NaOH), pH 11), acidic (0.1 M glycine-HCl, pH 2.5), or high salt (10 mM Hepes, 1 mM EDTA, 1 M NaCl, pH 7.4) buffer solutions under the same conditions, except that extraction was performed for 20 h with buffer exchanges after 1, 3, and 16 h.

## Co-immunoprecipitation

For co-immunoprecipitation experiments, 400 μL of magnetosomes with an $OD_{492}$ of 6 in buffer E (corresponding to 0.28 μg mL$^{-1}$ protein) were pelleted by centrifugation (15 min, 21,000 xg, 10 °C). The supernatant was carefully discarded, and the pellet was resuspended in 100 μL buffer 1 (10 mM HEPES, 1 mM EDTA, 50 mM NaCl, 0.1 mM PMSF, 0.5% (v/v) Triton X-100, pH 7.4). After incubation for 12 h at 4 °C on a roll mixer, samples were pelleted by centrifugation (15 min, 21,000 xg, 10 °C). Subsequently, 100 μL of the supernatant were transferred to new 1.5 mL reaction tube and 400 μL buffer 2 (10 mM HEPES, 1 mM EDTA, 50 mM NaCl, 0.1 mM PMSF, 0.1% (v/v) Triton X-100, pH 7.4) were added (input). Prior to immunoprecipitation, 25 μL of the GFP-Trap® Agarose (Chromotek, Planegg-Martinsried, Germany) affinity resin were equilibrated three times with 500 μL buffer 2, sedimented by centrifugation (5 min, 2,500 xg, 4 °C) and the supernatant was discarded. 450 μL of the diluted input were added to the equilibrated beads and after incubation for 1 h at 4 °C on a roll mixer, beads were sedimented by centrifugation (5 min, 2500 xg, 4 °C). The supernatant (unbound) and 50 μL of the unused input were diluted 1:1 with 2x SDS-sample buffer (100 mM Tris, 20% (v/v) glycerol (v/v), 4% (w/v) SDS, 0.2% (w/v) bromophenol blue, 200 mM DTT, pH 6.8) and stored at -20 °C until further analysis. The beads were washed three times with 500 μL buffer 2, sedimented by centrifugation (5 min, 2,500 xg, 4 °C) and the supernatant was discarded. Finally, the beads were transferred to new 1.5 mL reaction tubes, resuspended in 80 μL 2x SDS-sample buffer, boiled for 5 min at 95 °C to dissociate immunocomplexes (IP) from beads and sedimented by centrifugation (5 min, 2,500 xg, 10 °C). Samples were then evaluated via SDS-PAGE and subsequent Western immunoblots by loading 10 μL (input) and 20 μL (unbound, Co-IP) aliquots, respectively.

## Sample preparation for mass spectrometry

Magnetosome samples were prepared for mass spectrometry using filter aided sample preparation (FASP)[67]. Briefly, aliquots containing 100 μg of total protein were reduced with tris(2-carboxyethyl)phosphine followed by mixing with 200 μL of buffer UA (8 M urea in 0.1 M Tris/HCl, pH 8.5) and loading on a Microcon YM 30 (Merck-Millipore, Darmstadt, Germany) filtration device by centrifugation. Subsequently, proteins were alkylated by adding iodoacetamide in buffer UA and digested for 18 hours at 37 °C with trypsin at a ratio of 1:100 (trypsin:protein) in 40 μL 50 mM Tris/HCl. Peptides were eluted by centrifugation with 100 μL 50 mM Tris/HCl, twice. Eluates were pooled and resulting peptides were purified by Pierce C18 Tips 100 μL (Thermo Fisher Scientific, Waltham, USA). Therefore, C18 tips were wetted with 200 μL of 70% acetonitrile and equilibrated with 200 μL 3% acetonitrile. Peptides were bound by aspirating and dispensing ten times. Bound peptides were washed and eluted with water and 60% acetonitrile, respectively. Eluted peptides were dried and stored at -80 °C until further use.

## Mass spectrometry of magnetosomes

Purified peptides were reconstituted with 10 μL 0.1% acetic acid and analyzed by reversed phase liquid chromatography (LC) electrospray ionization (ESI) MS/MS using a QExactive Hybrid-Quadrupol-Orbitrap mass spectrometer (Thermo Fisher Scientific, Waltham, USA). In brief, nano reversed phase LC columns (20 cm length x 100 μm diameter) packed with 3.0 μm C18 particles (Dr. Maisch GmbH, Ammerbuch Entringen, Germany) were used to separate the purified peptides with an EASY nLC 1000 system (Thermo Fisher Scientific, Waltham, USA). The peptides were loaded with buffer A (0.1% acetic acid) and subsequently eluted by a non-linear gradient of 166 min from 2% to 99% buffer B (0.1% acetic acid, 99.9% acetonitrile) at a flow rate of 300 nL min$^{-1}$. A full scan was recorded in the Orbitrap with a resolution of 70,000. The twelve most abundant precursor ions were consecutively isolated by the quadrupole and fragmented via higher-energy collisional dissociation with a normalized collision energy of 27.5%. MS2 scans were recorded with a resolution of 17,500. Unassigned charge states, singly charged ions, as well as ions of charge 7 and higher were rejected and the lock mass correction was enabled.

Database searching and quantification was done with MaxQuant version 1.6.3.4[68]. with the published genome sequence of MSR-1 (GenBank: CP027526.1)[17]. The MaxQuant generic contaminants database was used. Database search was based on a strict tryptic digestion with two missed cleavages permitted. Carbamidomethylating on cysteine was considered as a fixed modification and oxidation of methionine as a variable modification. MaxQuant computed LFQ intensities were loaded into Perseus 1.6.2.2[69] and log$_2$ transformed. Putative contaminants, reverse hits, and proteins identified by site only were removed and a list containing proteins identified in all samples was exported to Excel. Mean log$_2$ differences and respective $P$-values were obtained by a two-sided two sample $t$-test over three biological replicates.

The mass spectrometry proteomics data have been deposited to the ProteomeXchange (http://proteomecentral.proteomexchange.org) consortium via the PRIDE partner repository[70] with the dataset identifier PXD032959.

## Epifluorescence microscopy

For epifluorescence microscopy, MSR-1 strains were grown in 3 mL FSM at 2% O2 at 28 °C without shaking for approximately 20 h. When required, gene expression was induced by addition of 100 ng mL$^{-1}$ anhydrotetracycline or 2 mM IPTG. For imaging, cells were immobilized on 1% agarose pads supplemented with FSM media components

(except for peptone and yeast extracts). Therefore, 3 μL of cell suspension were pipetted on agarose pads and covered with a coverslip. Samples were then imaged with an Olympus BX81 microscope equipped with a 100× UPLSAPO100XO objective (NA1.4), an OrcaER camera (Hamamatsu), and DIC contrast. Epifluorescence micrographs were recorded in Z-stacks with 750 ms exposure time per image and then deconvoluted employing 200 iterations of the Richardson-Lucy algorithm[71,72] using the DeconvolutionLab 2.0.0 plugin[73] the ImageJ Fiji package[74].

## Structured illumination microscopy

3D-SIM (striped illumination at 3 angles and 5 phases) was performed on an Eclipse Ti2-E N-SIM E fluorescence microscope (Nikon) equipped with a CFI SR Apo TIRF AC 100×H NA1.49 Oil objective lens, a hardware based 'perfect focus system' (Nikon), LU-N3-SIM laser unit (488/561/640 nm wavelength lasers) (Nikon), and an Orca Flash4.0 LT Plus 17 sCMOS camera (Hamamatsu). Calibration of the objective correction collar and SIM grating focus was performed with TetraSpeck fluorescent beads (T-7279 TetraSpeck microspheres). For 3D-SIM imaging, cells were prepared as described above except that 8-Well μ-Slides with 1.5H (170 ± 5 μm) D 263 M Schott glass bottom (ibidi GmbH, Gräfelfing, Germany) were used. 3D SIM z-series were acquired with 120 nm z-step spacing and exposure times in the range of 20 to 100 ms at 25 to 75% laser power. EM515/30 and EM595/31 filters and fluorescence excitation with 488 nm and 561 nm lasers were used for imaging of GFP and mCherry/mScarlet-I, respectively. Image reconstruction was performed in NIS-Elements 5.01 (Nikon) using the 'stack reconstruction' algorithm with the following parameter settings: The 'illumination modulation contrast' was set to 'auto'. The 'high resolution noise suppression' was set to 0.1.

## Transmission electron microscopy

For transmission electron microscopy (TEM) analysis, cells were grown at 28 °C under microaerobic conditions (2% $O_2$) over-night. One milliliter cell suspension (OD565 ~ 0.2-0.3) was then concentrated by centrifugation at 1.000 xg for 3 min, followed by resuspension in ~50 μL of residual medium. Afterwards cells were adsorbed onto carbon coated copper mesh grids (CF200-CU, Electron Microscopy Sciences, Pennsylvania) and washed with ddH$_2$O twice. Images were recorded using Zeiss EM 902 A and Jeol JEM-1400 Plus electron microscopes at an accelerating voltage of 80 kV. For data processing, interpretation, and analysis, the software packages DigitalMicrograph (Gatan) and the ImageJ Fiji package[74] were used. For determinations of magnetite particle numbers per cell, at least 57 cells were analyzed and at least 300 particles were measured for analysis of magnetite particle diameters.

Magnetosome chain formation was analyzed from TEM images as depicted in Supplementary Fig. 2. Briefly, TEM images were segmented using the ImageJ plugin Trainable Weka Segmentation v3.2.28[75] for extraction of cell boundaries and a difference of Gaussians (DoG) method (sigma1 = 1, sigma2 = 2)[76] to enhance the edges of the images and enable extraction of magnetosomes by thresholding. After manual curation to prevent erroneous segmentation of polyphosphate granules, binary magnetosome images were subjected to particle number analyses using the inbuilt "Analyze Particles" function of Fiji. For neighbor analyses, the "Neighbor Analysis" function of the Fiji Bio-Voxxel toolbox plugin 2.5.0[77] was used in particle neighborhood mode with 35 nm neighborhood radius. Intracellular magnetosome distribution maps were generated with MicrobeJ[78] from at least 593 particles and 16 cells.

## Bioinformatic analyses

For the detection of remote MFP homologies, Hidden Markov Model-based HHPred analyses were performed[18]. Using individual MFP sequences or MFP alignments as queries, the domain of unknown function DUF4870 (HOTT) and the Tic20 protein families were consistently the only hits with significant statistical support in various databases like PFAM[79], TIGRFAMs[80], or COG[81]. Subsequently, 5200 protein sequences of the identified HOTT and Tic20 protein families were retrieved from the InterPro database[82]. After an initial Cluster analysis of sequences (CLANS)[23], 229 HOTT and Tic20 family proteins of bacterial, archaeal and eukaryotic origin were selected to achieve a broad phylogenetic distribution. These sequences were again analyzed by CLANS using default settings for 150,000 iterations. Only if the $P$-value for a pair of sequences is less than $10^{-5}$ in the all-against-all BLAST search, the corresponding edges between nodes are shown as gray or black lines. To generate maximum-likelihood trees of MamF-like proteins or HOTT and TIC20 family proteins 59 sequences were aligned using MAFFT 7.474[83], respectively. Trimmed alignments (TrimAI 1.3[84], no gaps and gap threshold 0.7, respectively) were then used to infer maximum-likelihood trees with IQ-Tree 1.6.11[85] under the LG + G4 or mtInv+F + I + G4 models as suggested by ModelFinder[86]. Bootstrap support was derived by ultrafast bootstrap approximation with 1000 iterations. The phylogenetic trees were visualized and annotated by iTOL online tool[87].

To infer the phylogenetic distribution of the HOTT and Tic20 families, a total of 16840 rRNA sequences (16S or 18S) from organisms encoding these protein families were initially retrieved from the NCBI nucleotide database. After filtering for sequences of at least 1200 or 1655 nt length and removal of sequences with similarities above 97% (CD-Hit[88]), a total of 456 (HOTT) and 297 (Tic20) rRNA sequences were aligned using MAFFT 7.490[83], respectively. Trimmed alignments (TrimAI 1.3, no gaps) were then used to infer maximum-likelihood trees with IQ-Tree 1.6.12[85] under the GTR + F + R10 or TIM3 + F + R3 models as suggested by ModelFinder[86]. Bootstrap support was derived by ultrafast bootstrap approximation with 1000 iterations. The phylogenetic trees were visualized and annotated by iTOL[87]. All sequences and alignments were edited and analyzed using Geneious 8.1.4 (Biomatters, Auckland, New Zealand).

All sequences and alignments were edited and analyzed using Geneious 8.1.4 (Biomatters, Auckland, New Zealand).

For protein signal sequence predictions, SignalP 5.0[89] was used. Hydrophobicity analyses were performed using the grand average of hydropathy (GRAVY) calculator (http://www.gravy-calculator.de)[38,90].

Figures and RMSD values from pairwise structural comparisons using DALI[24] were generated for AlphaFold (AF) predictions[91,92] of Tic20/HOTT superfamily members, in comparison with the PDB structure of Tic20 (C. reinhardtii, PDB ID: 7xZI)[45]. All structural visualizations and comparisons were created using PYMOL[93].

## Statistics and reproducibility

All statistical analyses were performed with GraphPad Prism 7 software (GraphPad Software, Inc., La Jolla, CA, USA). All data were analyzed using two-tailed Student's t or Mann-Whitney U tests, respectively. A $P$-value of less than 0.05 was considered statistically significant. Further information about statistical details and methods is indicated in the figure legends, text, or methods. If not stated otherwise, values are given as mean ± standard deviation of the indicated sample size. Violin plots, bar plots, and intracellular magnetosome distribution maps were generated by Fit-o-Mat 0.752[94], Prism 7 software (GraphPad) and MicrobeJ[78], respectively. All experiments were performed at least twice.

## Reporting summary

Further information on research design is available in the Nature Portfolio Reporting Summary linked to this article.

# Data availability

All data supporting the findings of this study are included in the main text, the supplementary materials or the supplementary data files. Uncropped versions of blots/gels, swim halos in agar plates and bacterial two-hybrid assay agar plates are supplied in the Source data file

provided with this paper. The mass spectrometry proteomics data generated in this study have been deposited to the ProteomeXchange (http://proteomecentral.proteomexchange.org) consortium via the PRIDE partner repository[70] under the accession code PXD032959. The accession codes of structures and AlphaFold models used for analysis are 7XZI (Tic20); AF-A4U547-F1-model_v2 (MmsF, accession date 18.03.2022). Source data are provided with this paper.

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

## Acknowledgements

We would like to thank Gabriele Zaus and Ulrike Brandauer for technical assistance as well as Stephanie Bauer and Michael Gass for help during immunoblot analysis, two-hybrid assays, strain, and plasmid construction. The authors are grateful for financial support by the German Research Foundation (DFG, grants UE200/1-1 and INST 91/374-1 LAGG) (RU) and the Federal Ministry of Education and Research (BMBF, grant MagBioFab) (RU).

## Author contributions

Conceptualization and methodology: RU. Investigation: AP, FA, AS, TSc, HH and RU (conducted the biochemical and cell biological studies), TSu and DB (performed MS proteomics and analyzed MS data). Formal analysis: RU. Funding acquisition: RU. Supervision: RU. Writing - Original draft: AP and RU. Writing - Review & editing: all authors.

## Funding

## Competing interests

The authors declare no competing interests.
