## [Peer Review file · Nature Communications]

MamF-like proteins are distant Tic20 homologs involved in organelle assembly in bacteria

Corresponding Author: Dr René Uebe

Version 0:

Reviewer comments:

Reviewer #1

(Remarks to the Author)

The manuscript by Uebe et al, reports on very interesting discovery of magnetosome membrane proteins that share homology with plastid import component Tic20. By series of genetic and cellular experiments authors bring a collection of indirect evidences suggesting that these proteins collectively labeled as MFPs are involved in the integration of substrate polypeptides into the magnetosome membrane.

I am not a bacteriologist, so cannot judge how difficult these studies are to be done, but I appreciate the discovery and the conclusive sum of experiments that MFPs might be orthologues of Tic20. Of course, the decisive experiment showing direct defect in the substrate protein insertion or complementation of MFPs with Tic20 would bring clarity into the story but I know quite well that things are complicated when testing evolution in the tube. What I lack in the manuscript is better structure. The main message of the paper is spread across the main and supplementary data. So if the authors want to demonstrate that MFPs are orthologues of Tic20 they should get all these data to the main figure and simplify the text on the cytoskeletal data, morphology of magnetosomes and crystal formation. These are nice phenotypes but the main message is the homology and protein insertion into the magnetosome membrane. However, I really like the story and I recommend just to restructure the text and figures to become more focused on MFPs function. No more experiments.

Reviewer #2

(Remarks to the Author)

Manuscript: "Homologs of the plastidial preprotein translocase core component Tic20 mediate organelle assembly in bacteria" by Paulus A et al..

-The authors, report that MamF-like membrane proteins (MamF, MmsF and MmxF) are part of a superfamily that includes DUF4870 and Tic20 families, suggesting they might have a role in magnetosome assembly. Combining gene deletion and in vivo complementation they demonstrate that the concomitant deletion of all MamF-like proteins not only affects the magnetite size but also leads to loss of magnetosome chain formation, a phenotype reminiscent of the deletion of the cytoskeletal MamJ protein. Cell fractionation coupled to western blot analysis and quantitative proteomics demonstrated that in the absence of MamF-like proteins, the MamJ, MamY (less) and the magnetite size regulating proteins (MamD, Mms5 and MamR) are specifically reduced in magnetosome membranes, while their protein amounts in the cell remain unaffected. -MamJ was thought to be involved in the magnetosome anchoring to the cytoskeleton via protein-protein interaction with MmsF and MmxF. In a novel perspective in the field, the authors demonstrated that MamJ is an integral membrane protein, and its C-terminus is responsible for its proper targeting to the magnetosome membrane. The authors propose that MamJ membrane targeting and insertion is mediated by the MFPs who were previously proposed to only control the magnetite crystal growth.

We find that the research is well-conducted, the methodology is robust, with techniques relevant to the research question, and the data collection process is clearly outlined. The authors present their results clearly and conclusions are logical and supported by the data. Overall, the paper is well-structured and demonstrates a comprehensive understanding of the research area. We are of the opinion that the findings will significantly contribute to the field of magnetosome formation. However, we have the following concern with the manuscript: There is exaggerated emphasis on the sequence homology between the MFPs and the Tic20 proteins, to the extent that we feel that the title and abstract give the wrong impression about the content of results. The reviewer is of the opinion that the authors should focus on their magnetosome-significant

results (these deserve to have the first place in the manuscript) and discuss the found homology to the Tic20 families as an intriguing, nice discussion point that perhaps opens up future directions. We recommend that the authors revise the title, abstract, introduction and discussion to better reflect the manuscript's results and significance before publication.

Minor comments

- #1.
Introduction: MamF-like proteins should be discussed in the introduction.
- #2.
Fig. 1A and B: Nomenclature should be the same between panels. Now its HOTT(DUF4870) in B versus DUF4870 in A.
- #3.
Fig. 1B: Please introduce the "SF" subfamilies in fig. legend.
- #4.
Fig 1C:
 - a. The indication for panel "C" should be within the Panel C area, now is within the B area.
 - b. Purple and grey look very much alike. A lighter grey for the Tic20 structure would enable visual distinction.
 - c. Please indicate in fig legend what for stands the acronym IMH.
- #5.
Fig 2: Maintaining the same strain order in all panels (A-F), within colored categories, will enable the reader to follow results.
- #6.
Fig 2F:
 - a. In the legend, "l" should be "i" as in figure.
 - b. The representative swim halos do not always match the data in the graph. For example, in panel (i) WT and Δ mamF/mmxF show a significant difference yet in panel (ii) the corresponding swim halos look alike. How is that explained?
 - c. If mmsF complements the swim halo in Δ F3::mmsF, why Δ mamF/mmxF panel is not like wt (panel i)?
 - d. Results in panel (ii) refer to 72 hours of incubation and those of panel (i) to 24 hours (see legend). Why? Differences like those mentioned in comment b can be attributed to the time component in a meaningful way?
- #7.
page 3, lines 97-99:....."expression of each mamF-like gene within Δ F3 restored magnetosome crystal sizes at least partially, whereas magnetosome chain formation was reconstituted by expression of mmsF or mmxF but not mamF (Fig. 2A, C)."
 - a. Wouldn't it be more accurate to say that "some of the magnetosome features were corrected by expression of the single genes but never all of them by one gene complementation."...?
- #8.
Page 3, line 99:"chain formation was reconstituted by expression of mmsF or mmxF....."
This is clear for mmsF but not for the mmxF (2C ; 2E. Only in the Δ F3::mmsF strain the magnetosome chain formation is clearly restored, Δ F3::mamF show clustered (not chained) magnetosomes and Δ F3::mmxF show widely distributed magnetosomes (there is barely the beginning of a chain formation; certainly not like wt or Δ F3::mmsF).
- #9.
Fig 3A:
 - a. The authors discuss the reduced amounts of MamY and MamJ in the Δ F3 magnetosome extracts compared to WT. How is the significant increase of MamK in Δ F3 compared to WT explained by the authors?
 - b. The chain formation of Δ F3::6His-mmsF is not discussed in the main text. Does 6His-MmsF complement as well as MmsF in Fig. 2C? Also Δ F3::6His-mmsF exhibits over-targeting of MamJ and MamK to the magnetosome, and a new band appearing on MamY. How do the authors explain these phenotypes?
- #10.
Fig. 3B: Comparisons would benefit from adding the WT strain here.
- #11.
page 4, line 120: "This suggestion is supported by the stronger MamJ depletion, the phenotypic similarity of the mamJ deletion mutant with MFP mutant strains that still express mamF as the only MFP gene (Δ F3::mamF, Δ mmsF Δ mmxF) as well as the lack of MamJ in magnetosome membrane extracts of all MFP mutant strains that simultaneously lack mmsF and mmxF (Fig. 3C and Extended Data Fig. 3A)..... This sentence is not clear.
A simpler, shorter one might be better: The phenotypic similarities between Δ mamJ, Δ F3::mamF and Δ mmsF Δ mmxF combined with the observed absence of MamJ in magnetosome membranes from strains that lack both mmsF and mmxF (Fig. 3C and Extended Data Fig. 3A).....
- #12.
page 4 , line 128:....."MamK was again identified in equal quantities in WT and Δ F3 magnetosome membrane extracts (Fig.

3D)”

This protein is not indicated in Figure 3D.

#13.

Fig 4C: Please replace Buffer E” with HEPES or with nothing (-) if you used buffer E+NaCO₃ and buffer E+SDS to the next lanes. The buffer composition is important discriminator in this figure

#14.

Lines 203-205; Fig.4F: When removing 335-358 residues, the authors observed that MamJ can bind the cytoskeleton but cannot chain the magnetosome to the cytoskeleton, despite the gly-rich region being present. This phenotype is similar to removing half of the gly-rich region. Do the authors suggest that the membrane inserted region of MamJ is extended before the glycine-rich region or that an important linker region between the gly-rich integral membrane domain and the N-domain of the protein exists?

#15.

Page 6, line 214:”To discriminate between both possibilities, MmsF variants lacking all acidic (D34N, D36N, D37N, E38N) or charged (D34N, R35Q, D36N, D37N, E38N) residues of loop 1 were tested for their ability to increase magnetosome crystal sizes in $\Delta F3$ (Fig. 5A). Consistent with an indirect role during magnetite growth, both tested variants restored crystal sizes to the same level as wild-type MmsF (Fig. 5B”

To facilitate comparisons the wt, $\Delta F3$ and $\Delta F3::mmsF$ controls should be added in Fig. 5

#16.

Fig 5B: For the same strains, values slightly differ from Fig 5B to Fig 2A. For example, the magnetosome crystal size of $\Delta F3$ and $\Delta F3::mmsF$ is about 22 and 30 respectively in Fig 2A versus 19 and 25 respectively in Fig 5A. How is that justified?

#17.

Page 29: Panel legends B and C are reversed relative to the Figure panel indication (Fig. 5)].

#18.

“Deletion of all mamF-like genes in MSR, for example, resulted in the mislocalization of several magnetosome proteins (e.g. MamJ, MamD, Mms5), leading to magnetosome mispositioning, reduced magnetite biomineralization as well as loss of magnetotaxis (Fig. 2). Likewise, Tic20-I mutants of *Arabidopsis thaliana* demonstrated impaired plastid import of photosynthesis-related preproteins, resulting in abnormal chloroplast development and albino phenotypes 17,32. Thus, in both cases only a small subset of organellar proteins is affected.” [6 / 244] The authors link their observations on magnetosome mispositioning, with preproteins import in the Tic-20 apparatus. The connection appears a bit broad as no evidence for protein translocation is discussed in the magnetosome, rather coordination of proteins to assemble and localize an organelle.

#19.

Materials and Methods

a. page 8, line 320:... “modified flask standard medium”. Rather than sending to reference [50], it would be better to get the composition here.

b. It has been very difficult to find a specific buffer composition across the manuscript. A supplementary table listing all the buffers used in the study, in alphabetical order, and their composition would be helpful.

d. page 15, 637: Figure S2 should be described as extended Data Figure 2.

Reviewer #3

(Remarks to the Author)

Reviewer #4

(Remarks to the Author)

The manuscript “Homologs of the plastidial preprotein translocase core component Tic20 mediate organelle assembly in bacteria” describes the identification and characterization of mamF-like proteins, and showed that they influence magnetosome formation and chain alignment by regulating the localization of the proteins MamD, Mms5, and MamJ using different techniques. MamF-like proteins were thought to directly affect magnetosome crystal maturation, but here the authors showed that they regulate the membrane localization of the mineral-producing proteins MamD and Mms5 or the chain-organizing protein MamJ to indirectly control mineral production and magnetosome positioning.

This study contains many very carefully performed experiments that largely support the conclusion. Thus, the manuscript is a good addition to the literature on magnetotactic bacteria and helps to understand the mystery of magnetosome protein targeting. However, the manuscript could be further improved by addressing the following points:

Major concerns:

1. This manuscript mainly characterized the function of MamF-like proteins, and only a few bioinformatic analyses (I am not an expert on bioinformatics) showed the possible link between MamF-like proteins and Tic20. It is not clear whether MamF-like proteins are translocases. I think the title of this manuscript should directly mention the main mamF-like proteins studied in this work.
2. The text and figures of this manuscript could be improved.
The introduction consists of only two paragraphs, with the first paragraph being too long to be clearly understood. The paragraph between lines 178-205 is also very long.
In a printed A4 paper, the fluorescence and TEM images are too small, e.g. Figure 3b, 4d, 4f, and Extended Data Figure 3A, 3C, 4A, 4D, 5C.
3. This manuscript discusses the localization of magnetosome membrane proteins. Therefore I believe that the specific protein targeting pattern of MamD and Mms6 needs to be mentioned in the introduction. Since MamD and Mms6 are only localized to crystal-containing magnetosomes.
4. Lines 31 and 32, "this process is supposedly induced bya molecular crowding-like mechanism". The reference paper has only suggested a possible explanation for a protein crowding model for magnetosome membrane invagination, but has not proven it. So please be more cautious when describing this hypothesis.
5. Line 99, Mmx_f only partially complements the deletion phenotype, why?
6. Line 105 and Fig. 2F, the authors did not explain what the halo aspect ratio is and how the halo shape correlates with the magnetotaxis ability of MSR-1.
7. CcfM is an important protein that interacts with MamY and MamK and is involved in magnetosome chain organization in MSR-1, why was the interaction between CcfM and MamF-like proteins not tested?
8. Lines 137-138, based on the images of Extended Data Fig. 3c, the localization of MamD, Mms5, MamR, MamG in $\Delta F3::mamF$ is not restored as they showed dots instead of chains. Same for $\Delta F3::mmx_F$. These do not match the description in lines 137-138.
9. Lines 206-207. I am not convinced that MamJ inserts into the magnetosome membrane. How could a soluble protein insert into a membrane lipid bilayer?
10. For Figure 1c, MamF is not really functional for protein targeting of MamD and MamJ, why not compare MmsF and Mmx_f with Tic20 instead of MamF? And why does MamF have different functions than MmsF and Mmx_f?

Minor points:

1. Line 31, to be consistent with other publications and easier to understand, please leave the strain name as "MSR-1" instead of "MSR".
2. Fig. 1c, the letter "C" should be placed in the correct position. To make it easier for readers, please indicate that gray stands for Tic20 and purple for MamF.
3. Fig. 3a requires a loading control to show the even loading of the samples.
4. Fig. 4d, the TEM images are too small and the magnetosomes need to be pointed out.
5. Extended Data Figure 4B, the magnetosomes need to be pointed out.

Reviewer #5

(Remarks to the Author)

In this article by Paulus and colleagues, it is studied the function of the magnetosome MamF-like proteins (MFPs). They initially performed a bioinformatic study to show that these proteins belong to the HOTT family and are distantly related to Tic20 family that mediate organelle-specific protein targeting in cyanobacteria and eukaryotes. Molecular and genetic approaches allowed them to show that the MFPs are essential for the regulation of magnetosome crystal size and chain formation. However, by studying the other magnetosome proteins interacting with MFPs such as MamJ, they show that MFPs indirectly regulate organelle positioning and magnetosome crystal size by mediating correct organelle assembly. The study is very interesting not only for the community working on biomineralization and magnetotactic bacteria but also extend to researcher working in prokaryotic and eukaryotic organelles. Thus this justify the publication of such study in a generalist journal. I have only very minor comments as I found the study very well performed with all required controls

carried out.

Line 57, I would remove « Recently » as the paper cited is from 2018.

Line 161, remove « indeed »

Version 1:

Reviewer comments:

Reviewer #2

(Remarks to the Author)

Revised manuscript: "Homologs of the plastidal preprotein translocase core component Tic20 mediate organelle assembly in bacteria" by Paulus A et al..

Following careful reading of the revised manuscript as well as the author's responses to reviewer's comments we conclude that the authors addressed most (if not all) our comments and we recommend it for publication.

Minor comments:

#1

Page 2 / Line 56, the acronym MTB is used, but not defined.

#2

The author's response to our previous comments #6.c and 6.d could be summarized and implemented in discussion.

#3

In response to our previous comment #8, the authors updated the figure with a new TEM micrograph to emphasize their statement. For transparency reasons, the authors could add the previous Fig. 2b, e panels in the Supplement to indicate some variation.

#4

In their reply to our previous comment #9a, about the increase of MamK in the magnetosome fraction, the authors suggest that there might be another binding partner. This might be an interesting point for the discussion.

Reviewer #3

(Remarks to the Author)

Reviewer #4

(Remarks to the Author)

The authors revised well to the comments and made the necessary corrections to the text and figures.

Dear reviewers,

please find our point-by-point responses to the reviewers' comments below with original comments/remarks in black and our answers in blue fonts:

Reviewer #1 (Remarks to the Author):

The manuscript by Uebe *et al*, reports on very interesting discovery of magnetosome membrane proteins that share homology with plastid import component Tic20. By series of genetic and cellular experiments authors bring a collection of indirect evidences suggesting that these proteins collectively labeled as MFPs are involved in the integration of substrate polypeptides into the magnetosome membrane.

I am not a bacteriologist, so cannot judge how difficult these studies are to be done, but I appreciate the discovery and the conclusive sum of experiments that MFPs might be orthologues of Tic20. Of course, the decisive experiment showing direct defect in the substrate protein insertion or complementation of MFPs with Tic20 would bring clarity into the story but I know quite well that things are complicated when testing evolution in the tube. What I lack in the manuscript is better structure. The main message of the paper is spread across the main and supplementary data. So if the authors want to demonstrate that MFPs are orthologues of Tic20 they should get all these data to the main figure and simplify the text on the cytoskeletal data, morphology of magnetosomes and crystal formation. These are nice phenotypes but the main message is the homology and protein insertion into the magnetosome membrane. However, I really like the story and I recommend just to restructure the text and figures to become more focused on MFPs function. No more experiments.

Thank you very much for your supportive comments and helpful advice to improve our manuscript and highlight the homology between MFPs and Tic20.

Based on the recommendation of the editor and comments of the other reviewers, we decided to give the magnetosome-specific results the main space in the manuscript.

Reviewer #2 (Remarks to the Author):

Manuscript: “Homologs of the plastidal preprotein translocase core component Tic20 mediate organelle assembly in bacteria” by Paulus A et al..

-The authors, report that MamF-like membrane proteins (MamF, MmsF and MmxF) are part of a superfamily that includes DUF4870 and Tic20 families, suggesting they might have a role in magnetosome assembly. Combining gene deletion and in vivo complementation they demonstrate that the concomitant deletion of all MamF-like proteins not only affects the magnetite size but also leads to loss of magnetosome chain formation, a phenotype reminiscent of the deletion of the cytoskeletal MamJ protein. Cell fractionation coupled to western blot analysis and quantitative proteomics demonstrated that in the absence of MamF-like proteins, the MamJ, MamY (less) and the magnetite size regulating proteins (MamD, Mms5 and MamR) are specifically reduced in magnetosome membranes, while their protein amounts in the cell remain unaffected.

-MamJ was thought to be involved in the magnetosome anchoring to the cytoskeleton via protein-protein interaction with MmsF and MmxF. In a novel perspective in the field, the authors demonstrated that MamJ is an integral membrane protein, and its C-terminus is responsible for its proper targeting to the magnetosome membrane. The authors propose that MamJ membrane targeting and insertion is mediated by the MFPs who were previously proposed to only control the magnetite crystal growth.

We find that the research is well-conducted, the methodology is robust, with techniques relevant to the research question, and the data collection process is clearly outlined. The authors present their results clearly and conclusions are logical and supported by the data. Overall, the paper is well-structured and demonstrates a comprehensive understanding of the research area. We are of the opinion that the findings will significantly contribute to the field of magnetosome formation. However, we have the following concern with the manuscript: There is exaggerated emphasis on the sequence homology between the MFPs and the Tic20 proteins, to the extent that we feel that the title and abstract give the wrong impression about the content of results. The reviewer is of the opinion that the authors should focus on their magnetosome-significant results (these deserve to have the first place in the manuscript) and discuss the found homology to the Tic20 families as an intriguing, nice discussion point that perhaps opens up future directions. We recommend that the authors revise the title, abstract, introduction and discussion to better reflect the manuscript's results and significance before publication.

Thank you very much for your careful evaluation, supportive comments, insightful questions and constructive suggestions. Based on these comments, we modified the manuscript by changing the title and rewriting the abstract, the introduction as well as the discussion of our manuscript (see the subsequent point-by-point responses).

Minor comments

#1. Introduction: MamF-like proteins should be discussed in the introduction.

While rewriting the introduction, we also added a sentence about the (previously known) putative functions of MFPs to the introduction.

“The small integral protein MamF and its paralogue MmsF, for example, are thought to promote magnetite crystal growth by direct interaction with the magnetite crystal surface or metal ions via a cluster of lumen-residing conserved acidic amino acids¹²⁻¹⁴.” (p.1-2, lines 46-49)

#2. Fig.1A and B: Nomenclature should be the same between panels. Now its HOTT(DUF4870) in B versus DUF4870 in A.

*Thank you for noting this inaccuracy. The nomenclature has now been unified within **Fig.1a and b** and the corresponding figure legend:*

“... the HOTT (DUF4870) and Tic20 protein families.” (p.22, line 975-976)

After its initial introduction as HOTT (DUF4870), the previously named DUF4870 family will henceforth be referred to solely as HOTT.

#3. Fig. 1B: Please introduce the “SF” subfamilies in fig. legend.

Thank you for pointing this out. We added the abbreviation for “SF” to the figure legend:

“The HOTT family includes subfamilies (SF) 1-6 and the MFPs.” (p.22, lines 978-979)

#4. Fig 1C:

a. The indication for panel “C” should be within the Panel C area, now is within the B area.

*Thank you for noting the mispositioning of the label. We have corrected its position in **Fig. 1**.*

b. Purple and grey look very much alike. A lighter grey for the Tic20 structure would enable visual distinction.

*We have changed the color in **Fig. 1c** according to your recommendations to visualize the different structures more clearly.*

c. Please indicate in fig legend what for stands the acronym IMH.

Thank you for highlighting the missing definition of the IMH abbreviation. We now introduce this abbreviation:

“Remarkably, despite only remote sequence homologies, multiple sequence alignments of HOTT and Tic20 family proteins, and structural superimposition using distance matrix alignment (DALI) analyses 24 revealed similar architectures, including a reentrant-like integral membrane helix (**IMH**) that is followed by a conserved charged loop and two additional integral membrane helices (Fig. 1c, d).” (p.2, lines 85-89)

#5. Fig 2: Maintaining the same strain order in all panels (A-F), within colored categories, will enable the reader to follow results.

*Thank you for the helpful comment. The order of the strains has been unified throughout all panels of **Fig. 2**.*

#6.Fig 2F:

a. In the legend, “I” should be “i” as in figure.

To avoid any confusion, we changed the labeling of Fig. 2Fi and Fii to Fig. 2f and g, respectively.

b. The representative swim halos do not always match the data in the graph. For example, in panel (i) WT and $\Delta mamF/mmxF$ show a significant difference yet in panel (ii) the corresponding swim halos look alike. How is that explained?

Thank you for pointing out this discrepancy. We repeated the experiment and indeed found that strain $\Delta mamF/mmxF$ shows differently shaped swarm halos. We apologize for the wrong image of the previous version of the manuscript and replaced it with an image from the repeated experiment. Further differences are only of minor nature which can be attributed to the longer incubation times (see comments below)

c. If *mmsF* complements the swim halo in $\Delta F3::mmsF$, why $\Delta mamF/mmxF$ panel is not like wt (panel i)?

*This is indeed an interesting side observation. In general, the swarm halo assay is influenced by the magnetite crystal size (which correlates with the magnetic moment of a particle) but is also highly sensitive towards changes in magnetosome chain organization (doi:10.1128/AEM.01976-19). As shown in Fig. 2a, strain $\Delta F3::mmsF$ possesses slightly larger particles than strain $\Delta mamF/mmxF$. Although both strains produce MmsF as the only MFP, *mmsF* expression in $\Delta F3::mmsF$ is higher than in $\Delta mamF/mmxF$ due to the use of a stronger promoter (P_{mamG} vs. the natural P_{mms6} ; see 10.1128/mSystems.00893-21). Similar to the virtually isogenic strain $\Delta F3::6His-mmsF$, the higher *mmsF* expression most likely leads to a better targeting of magnetosome proteins in $\Delta F3::mmsF$ (see Fig. 3a). While the improved targeting of biomineralization proteins (e.g. MamD, Mms5) leads to larger magnetite (i.e. more magnetic) crystals, increased magnetoskeletal protein levels cause the formation of slightly more cohesive magnetosome chains and lower numbers of “free” magnetosomes (Fig. 2b).*

Thus, due to smaller magnetosomes and loose magnetosome chains, strain $\Delta mamF/mmxF$ shows lower swarm halo aspect ratios than the WT. Strain $\Delta F3::mmsF$, on the other hand, shows increased swarm halo aspect ratios compared to strain $\Delta mamF/mmxF$ due to its larger (more magnetic) crystals and slightly more coherent magnetosome chains.

Although interesting, it would be beyond the scope of the current study to determine the relative contributions of the individual influencing factors. We therefore modified the text only slightly to acknowledge the main factors that contribute to the high swarm halo aspect ratios in $\Delta F3::mmsF$:

“In contrast, strains with coherent magnetosome chains and larger crystals show strong magnetic alignments (WT and $\Delta F3::mmsF$) resulting in swarm halos strongly elongated in the direction of the magnetic field (Fig. 2f, g).” (p.3, lines 121-123)

d. Results in panel (ii) refer to 72 hours of incubation and those of panel (i) to 24 hours (see legend). Why? Differences like those mentioned in comment b can be attributed to the time component in a meaningful way?

We chose to display swarm halos after 72 h in Fig. 2g (previously Fig. 2 Fii) for several reasons. On the one hand, the assay is growth dependent. Starting from low inocula, relatively faint swarm halos are visible after 24h. While these can be analyzed quantitatively, they are more difficult to visualize than swarm halos after 72 h when much higher cell densities are reached. On the other hand, we found slightly higher aspect ratios at 24 h than after 72 h. We assume this difference is the result of stronger nutrient depletion within the agar at later time points. These might result in different growth rates or additional tactic responses that superimpose onto the magnetotactic response. Therefore, we decided to provide the 24 h data in in Fig 2f (previously Fig. 2Fi).

#7. page 3, lines 97-99:.....“expression of each *mamF*-like gene within $\Delta F3$ restored magnetosome crystal sizes at least partially, whereas magnetosome chain formation was reconstituted by expression of *mmsF* or *mmxF* but not *mamF* (Fig. 2A, C).”

a. Wouldn't it be more accurate to say that “some of the magnetosome features were corrected by expression of the single genes but never all of them by one gene complementation.”...?

*Thank you for your comment. In our opinion, expression of *mmsF* is able to restore all magnetosome features in $\Delta F3$. In strain $\Delta F3::mmsF$, the magnetosome crystal size is restored to the level of the double deletion mutants while magnetosome chain formation and magnetotaxis are even improved. Unfortunately, *mmxF* expression complements magnetosome size and chain formation only partially. We assume that the partial complementation may derive from an unbalanced production of MmxF or an impaired folding due to the use of a much stronger, non-natural promoter (P_{mamG}). In contrast, *mamF* expression in $\Delta F3$ “fully” restores magnetosome crystal sizes but fails to do so for magnetosome chain formation as the MamF protein is unable to interact with MamJ.*

To clarify these differences more clearly, we modified the text:

*“Individual expression of *mmsF* or *mamF* within $\Delta F3$ restored magnetosome crystal sizes to the level of double deletion mutants whereas complementation with *mmxF* restored crystal sizes incompletely. In contrast, magnetosome chain formation was reconstituted by expression of *mmsF* and to some extent *mmxF* but not *mamF* indicating that the functions of the MFPs overlap only partially (Fig. 2a, c).” (p.3, lines 107-111)*

#8. Page 3, line 99:“chain formation was reconstituted by expression of *mmsF* or *mmxF*....”

This is clear for *mmsF* but not for the *mmxF* (2C ; 2E. Only in the $\Delta F3::mmsF$ strain the magnetosome chain formation is clearly restored, $\Delta F3::mamF$ show clustered (not chained) magnetosomes and $\Delta F3::mmxF$ show widely distributed magnetosomes (there is barely the beginning of a chain formation; certainly not like wt or $\Delta F3::mmsF$).

*Thank you for your comment. As described above, complementation of $\Delta F3$ with *mmxF* only partially restores the deletion phenotypes of the $\Delta F3$ mutant. This partial complementation leads to some degree of phenotypic heterogeneity with many cells having very short magnetosome chains (e.g. as shown in our initial manuscript in Fig. 2B and E) and few cells with longer magnetosome chains. As we intended to display only representative cells, we provided a TEM image of a cell with a short*

*magnetosome chain in the first version of our manuscript. However, to show that MmxF is able to restore magnetosome chain formation more convincingly, we now provide a new $\Delta F3::mxF$ TEM micrograph and segmented magnetosome chain from a cell with a more pronounced magnetosome chain in **Fig. 2b and e**.*

*Moreover, we want to point out that even the short magnetosome chains within $\Delta F3::mxF$ already lead to an at least slightly improved magnetotactic alignment in the swarm halo assay (**Fig. 2f, g**). Without this rudimentary chain formation one would not observe any magnetic alignment (see $\Delta F3$ and $\Delta F3::mamF$ in **Fig. 2f, g** for comparison).*

#9. Fig 3A: a. The authors discuss the reduced amounts of MamY and MamJ in the $\Delta F3$ magnetosome extracts compared to WT. How is the significant increase of MamK in $\Delta F3$ compared to WT explained by the authors?

Thank you for your comment. Overall, we are puzzled by the observation that MamK is present on the MM of strain $\Delta F3$ at all. In the almost complete absence of its "binding partner", MamJ, we would have also expected reduced MamK levels. Our findings therefore indicate that MamK must interact with additional magnetosome proteins besides MamJ. However, to our knowledge, no such interaction has been described yet. We can thus only speculate that a protein with an unaffected or slightly increased abundance in the $\Delta F3$ MM must also be able to weakly (as there is no chain formation) bind MamK.

While our quantitative proteomic analyses revealed no significant increase of the MamK level within the $\Delta F3$ magnetosome fraction, we modified the text to refer to the increased level of MamK in the Westernblot analyses:

*"..., while MamK was detectable at levels slightly higher than those of the WT (**Fig. 3a**)."* (p.3, lines 134-135)

b. The chain formation of $\Delta F3::6His-mmsF$ is not discussed in the main text. Does 6His-MmsF complement as well as MmsF in Fig. 2C? Also $\Delta F3::6His-mmsF$ exhibits over-targeting of MamJ and MamK to the magnetosome, and a new band appearing on MamY. How do the authors explain these phenotypes?

*Thank you for drawing our attention to the missing introduction of the $\Delta F3::6His-mmsF$ strain. This strain indeed phenocopies the $\Delta F3::mmsF$ strain. We therefore added a corresponding TEM image to **Supplementary Fig. 3a** and modified the text to introduce the new strain (see below). Over-targeting of the magnetoskeletal proteins in this strain is most likely based on the increased *mmsF* expression due to use of the strong *mamG* promoter (10.1128/mSystems.00893-21).*

*"Notably, expression of functional *His₆-mmsF* from the strong *mamG* promoter²⁶ (**Supplementary Fig. 3a**) mediated MM-targeting of all magnetoskeletal proteins in $\Delta F3$ at elevated levels." (p.3, lines 135-137)*

*The additional MamY band identified by the reviewer is not specific to the $\Delta F3::mmsF$ strain as it can also be detected in the MM fractions of the WT and $\Delta F3$, although at a lower level (see **Fig. 3a**). Since this band is not detectable in whole-cell extracts, it seems to be specific to magnetosomes. The reduced electrophoretic mobility of the additional band indicates that MamY might be modified posttranslationally. However, no posttranslational modifications have been described for MamY yet. Since magnetosome targeting of MamY is only indirectly affected by the deletion of MFPs (see*

10.1038/s41564-019-0512-8), we consider the discussion of a potential posttranslational modification of MamY as beyond the scope of this study.

#10. Fig. 3B: Comparisons would benefit from adding the WT strain here.

We agree with the reviewer's comment. We now included fluorescence micrographs from WT or complemented mutant strains in Fig. 3b.

#11. page 4, line 120:“This suggestion is supported by the stronger MamJ depletion, the phenotypic similarity of the mamJ deletion mutant with MFP mutant strains that still express mamF as the only MFP gene ($\Delta F3::mamF$, $\Delta mmsF\Delta mmxF$) as well as the lack of MamJ in magnetosome membrane extracts of all MFP mutant strains that simultaneously lack mmsF and mmxF (Fig. 3C and Extended Data Fig. 3A)“ This sentence is not clear.

A simpler, shorter one might be better: The phenotypic similarities between $\Delta mamJ$, $\Delta F3::mamF$ and $\Delta mmsF\Delta mmxF$ combined with the observed absence of MamJ in magnetosome membranes from strains that lack both mmsF and mmxF (Fig. 3C and Extended Data Fig. 3A).....

Thank you for your very helpful feedback. We modified the corresponding sentence according to your suggestion:

“This suggestion is supported by the stronger MamJ-depletion, the phenotypic similarities between $\Delta mamJ$, $\Delta F3::mamF$, and $\Delta mmsF\Delta mmxF$ and the absence of MamJ in MMs from strains that lack both mmsF and mmxF (Fig. 3c and Supplementary Fig. 3a).” (p.4, lines 140-142)

#12. page 4 , line 128:.....“MamK was again identified in equal quantities in WT and $\Delta F3$ magnetosome membrane extracts (Fig. 3D)”

This protein is not indicated in Figure 3D.

Thank you for pointing out the missing label for MamK. We now added a dash to indicate the proteomic results for MamK in Fig. 3d.

#13. Fig 4C: Please replace Buffer E” with HEPES or with nothing (-) if you used buffer E+NaCO₃ and buffer E+SDS to the next lanes. The buffer composition is important discriminator in this figure

Thank you for your comment. Buffer E consists of 10 mM HEPES and 1 mM EDTA. Therefore, in our opinion, changing the label to HEPES would be misleading. Use of a (-) label, on the other hand, would also not entirely reflect the composition of the used buffers as the NaCO₃-treated sample did not contain buffer E to reach a pH of 11.3. To clarify the composition of the buffers during the different treatments, we now labeled the samples with:

“buffer E; NaCO₃; and buffer E + SDS, respectively.” in Fig. 4c and the corresponding figure legend:

“...buffer E- (negative control), carbonate- (NaCO₃), and SDS-treated (buffer E+SDS, positive control),...” (p.24, lines 1059-1060)

#14. Lines 203-205; Fig.4F: When removing 335-358 residues, the authors observed that MamJ can bind the cytoskeleton but cannot chain the magnetosome to the cytoskeleton, despite the gly-rich region being present. This phenotype is similar to removing half of the gly-rich region. Do the authors suggest that the membrane inserted region of MamJ is extended before the glycine-rich region or that an important linker region between the gly-rich integral membrane domain and the N-domain of the protein exists?

Thank you for pointing out this interesting question. We suggest that deletion of the MamJ residues 335-358 interferes with the accessibility or folding of the putative IMH. We therefore added the following sentence to the discussion:

“Only truncations that partially deleted the putative IMH (D378-426) or potentially interfered with its accessibility or folding (D335-358) disrupted magnetosome chain reconstitution by MamJ, likely by preventing its MM binding (Fig. 4d-f and Supplementary Fig. 4d, e).” (p.6, lines 272-276)

#15. Page 6, line 214:”To discriminate between both possibilities, MmsF variants lacking all acidic (D34N, D36N, D37N, E38N) or charged (D34N, R35Q, D36N, D37N, E38N) residues of loop 1 were tested for their ability to increase magnetosome crystal sizes in $\Delta F3$ (Fig. 5A). Consistent with an indirect role during magnetite growth, both tested variants restored crystal sizes to the same level as wild-type MmsF (Fig. 5B”

To facilitate comparisons the wt, $\Delta F3$ and $\Delta F3::mmsF$ controls should be added in Fig. 5

Thank you for the important comment. Within Fig. 5a, we now provide TEM images of the corresponding negative ($\Delta F3$) and positive ($\Delta F3::mmsF$) controls. However, as the experiment aims to assess the potential of different mmsF alleles to restore magnetosome biomineralization and chain formation within $\Delta F3$, the WT was excluded from the analyses because it would not contribute to the conclusion.

#16. Fig 5B: For the same strains, values slightly differ from Fig 5B to Fig 2A. For example, the magnetosome crystal size of $\Delta F3$ and $\Delta F3::mmsF$ is about 22 and 30 respectively in Fig 2A versus 19 and 25 respectively in Fig 5A. How is that justified?

The experiments for Fig. 2 and Fig. 5 were performed independent of each other. We therefore believe that small differences in cultivation or fluctuation of the medium composition caused the observed differences in the magnetosome crystal sizes within the same strains. However, this does not compromise the conclusions from these experiments as the corresponding control strains were cultivated under identical conditions.

#17. Page 29: Panel legends B and C are reversed relative to the Figure panel indication (Fig. 5).

Thank you for pointing out the incorrect labeling. In the revised manuscript, we corrected this mistake in the legend for Fig. 5:

“...b Violin plot showing the magnetite crystal size distribution of strain $\Delta F3$ expressing WT and mutant mmsF variants as well as strains $\Delta A13\Delta mms5/mmxF/mamR$ and $\Delta A13\Delta mms5/mmxF/mamR::mmsF$. The number of analyzed particles [N] is indicated. The min, max, and mean values are given by bars. Statistical significance was estimated using an unpaired two-tailed Mann-Whitney U test (, P-value ≤ 0.05 ; **, P-value ≤ 0.01 ; ***, P-value < 0.001 ; ****, P-*

value < 0.0001; ns, not significant (P ≥ 0.05)). Raw data is provided in Supplementary Data Table 5. c Representative TEM micrographs of the $\Delta A13\Delta mms5/mmxF/mamR$ mutant and the complemented strain $\Delta A13\Delta mms5/mmxF/mamR::mmsF$. The positions of the small magnetosomes are indicated by arrows. (Scale bars: 500 nm)... (p.25, lines 1082-1090)

#18. “Deletion of all *mamF*-like genes in MSR, for example, resulted in the mislocalization of several magnetosome proteins (e.g. MamJ, MamD, Mms5), leading to magnetosome mispositioning, reduced magnetite biomineralization as well as loss of magnetotaxis (Fig. 2). Likewise, Tic20-I mutants of *Arabidopsis thaliana* demonstrated impaired plastidal import of photosynthesis-related preproteins, resulting in abnormal chloroplast development and albino phenotypes 17,32. Thus, in both cases only a small subset of organellar proteins is affected.” [6 / 244] The authors link their observations on magnetosome mispositioning, with preproteins import in the Tic-20 apparatus. The connection appears a bit broad as no evidence for protein translocation is discussed in the magnetosome, rather coordination of proteins to assemble and localize an organelle.

We agree with the reviewer's comment and removed the direct comparison between Tic20 and MFP deletion phenotypes as it might be too specific. Nevertheless, our bioinformatic analyses consistently and reproducibly showed that MFP belong to the HOTT family that is related to the Tic20 protein family. In the absence of any characterized member within the HOTT family we find it legitimate to compare MFPs to Tic20 (the only characterized member of the whole TIC20/HOTT superfamily) to discuss possible roles of the MFPs. We are aware of the distant homology and therefore modified our manuscript to express our conclusion more carefully: e.g. “Based on this supposed Tic20 function and the ability of MFPs to assemble into larger oligomers¹⁴, it is tempting to speculate that MFPs might form channels that facilitate the assembly of magnetosomes by inserting proteins into the MM that are not recognized by generic membrane protein integrases (e.g. the Sec translocon).” (p.7, lines 306-309)

#19. Materials and Methods

a. page 8, line 320:... “modified flask standard medium”. Rather than sending to reference [50], it would be better to get the composition here.

We agree with the reviewer's comment. We now provide the composition within the modified manuscript:

“Unless stated otherwise, MSR-1 strains were grown microaerobically at 28 °C in modified flask standard medium (FSM, pH 7, 10 mM HEPES, 12 mM potassium lactate, 4 mM NaNO₃, 0.74 mM KH₂PO₄, 0.60 mM MgSO₄ · 7 H₂O, 50 μM Fe(III)-citrate, 0.01% (w/v) yeast extract, 0.3% (w/v) soybean peptone) with moderate agitation (120 rpm).” (p.8, lines 353-357)

b. It has been very difficult to find a specific buffer composition across the manuscript. A supplementary table listing all the buffers used in the study, in alphabetical order, and their composition would be helpful.

*We agree with the reviewer's comment. Buffer compositions are now provided within **Supplementary Table 4***

d. page 15, 637: Figure S2 should be described as extended Data Figure 2.

*We unified the wording for our supplementary data e.g. “**Supplementary Fig. 2**” throughout the manuscript.*

Reviewer #3 (Remarks to the Author):

We greatly appreciate your effort and valuable contributions in co-reviewing this manuscript.

Reviewer #4 (Remarks to the Author):

The manuscript “Homologs of the plastidal preprotein translocase core component Tic20 mediate organelle assembly in bacteria” describes the identification and characterization of mamF-like proteins, and showed that they influence magnetosome formation and chain alignment by regulating the localization of the proteins MamD, Mms5, and MamJ using different techniques. MamF-like proteins were thought to directly affect magnetosome crystal maturation, but here the authors showed that they regulate the membrane localization of the mineral-producing proteins MamD and Mms5 or the chain-organizing protein MamJ to indirectly control mineral production and magnetosome positioning.

This study contains many very carefully performed experiments that largely support the conclusion. Thus, the manuscript is a good addition to the literature on magnetotactic bacteria and helps to understand the mystery of magnetosome protein targeting. However, the manuscript could be further improved by addressing the following points:

Major concerns:

1. This manuscript mainly characterized the function of MamF-like proteins, and only a few bioinformatic analyses (I am not an expert on bioinformatics) showed the possible link between MamF-like proteins and Tic20. It is not clear whether MamF-like proteins are translocases. I think the title of this manuscript should directly mention the main mamF-like proteins studied in this work.

Thank you very much for your positive comments, insightful questions and helpful advice to improve our manuscript. MamF-like proteins have now been included within the title. Additionally, we also removed the reference to the core role of Tic20 within the plastidal TIC preprotein translocase: “MamF-like proteins are distant Tic20 homologs involved in organelle assembly in bacteria” (p.1, line 2)

While Tic20 plays an important role for plastidal preprotein import (e.g. 10.1111/j.1365-313X.2011.04551.x;), it is currently under debate if it functions as a protein-transducing channel subunit within the Tic complex. To emphasize this debate, we have also revised our discussion and articulated our conclusions more carefully.

*Beyond these changes, we want to highlight that our bioinformatic analyses included six different methods that all **independently** revealed the homology between the HOTT (including MFPs) and the Tic20 protein families (**Fig. 1 and Supplementary Fig. 1**). Compared to other studies that suggest distant homologies between proteins or protein families, our work thus utilizes a significantly greater number of diverse methods (e.g. 10.1038/nsmb.2261; 10.1016/j.cub.2011.08.060; 10.1016/j.cell.2021.05.041). Nevertheless, to further strengthen our findings, we now included Distance-matrix ALIGNment (DALI) analyses (10.1002/pro.3749) to enable an unbiased and sensitive comparison of protein structure similarities (**Fig. 1c**).*

2. The text and figures of this manuscript could be improved.

In agreement with the suggestions of all reviewers we restructured the text and modified the figures to enhance clarity (see comments above).

The introduction consists of only two paragraphs, with the first paragraph being too long to be clearly understood.

In agreement with the comments of reviewers 2 and 4, we revised the structure of the introduction. We therefore have shortened the first part to enhance clarity and now also describe the MFPs within the introduction.

“The small integral protein MamF and its paralogue MmsF, for example, are thought to promote magnetite crystal growth by direct interaction with the magnetite crystal surface or metal ions via a cluster of lumen-residing conserved acidic amino acids¹²⁻¹⁴.” (p.1-2, lines 46-49)

The paragraph between lines 178-205 is also very long.

We acknowledge that this paragraph is quite dense and references numerous experiments and results, which may give the impression of being rather lengthy. However, we believe that shortening the paragraph would be counterproductive, as it might compromise the clarity and comprehension of the data.

In a printed A4 paper, the fluorescence and TEM images are too small, e.g. Figure 3b, 4d, 4f, and Extended Data Figure 3A, 3C, 4A, 4D, 5C.

*We agree with the reviewer and increased the size of the mentioned images. In **Fig. 4 and Supplementary Fig. 4**, we additionally provide magnified images of the magnetosomes.*

3. This manuscript discusses the localization of magnetosome membrane proteins. Therefore I believe that the specific protein targeting pattern of MamD and Mms6 needs to be mentioned in the introduction. Since MamD and Mms6 are only localized to crystal-containing magnetosomes.

Thank you for pointing out this important point. In the revised manuscript, we now introduced these proteins and their targeting mode within the introduction:

“Some magnetosome proteins, however, follow different targeting routes. Mms6 and MamD (also known as Mms7), for example, are translocated into preexisting magnetosomes in a folded state upon induction of magnetite biomineralization^{9,10}.” (p.1, lines 39-42)

4. Lines 31 and 32, “this process is supposedly induced bya molecular crowding-like mechanism”. The reference paper has only suggested a possible explanation for a protein crowding model for magnetosome membrane invagination, but has not proven it. So please be more cautious when describing this hypothesis.

To emphasize more clearly that the magnetosome biogenesis model remained unproven and is currently only a hypothesis, we have refined the sentence:

“It has been hypothesized that in *Magnetospirillum gryphiswaldense* MSR-1 and related Alphaproteobacteria, this process is induced by the assembly of magnetosome-specific protein complexes within the cytoplasmic membrane that initiate magnetosome membrane (MM) formation through a molecular crowding-like membrane-bending mechanism⁸.” (p.1, lines 34-37)

5. Line 99, MmxF only partially complements the deletion phenotype, why?

Partial complementation of Magnetospirillum mutants is a well-known phenomenon (e.g. 10.1111/mmi.12683; 10.1111/mmi.12317; 10.1093/nar/gkad1230). Unfortunately, the origin of the partial $\Delta F3$ complementation remained unknown in most cases. We can thus only speculate that the partial $\Delta F3$ complementation may result from an unbalanced production of MmxF or an impaired folding as MmxF might be sensitive to high expression levels driven by the use of a much stronger, non-natural promoter (P_{mamG}). To clarify this, we included a corresponding sentence within the figure legend of Fig. 2a:

*“The partial complementation of $\Delta F3$ with *mmxF* likely derives from the use of the strong but non-natural *mamG* promoter.” (p.23, lines 1002-1003)*

6. Line 105 and Fig. 2F, the authors did not explain what the halo aspect ratio is and how the halo shape correlates with the magnetotaxis ability of MSR-1.

Thank you for bringing to our attention that the swarm halo assay requires further clarification. We therefore modified the text to explain how the swim halo aspect ratio correlates with magnetotaxis (and how it is determined):

“Consequently, MFP mutants with reduced crystal sizes and defective magnetosome chains are severely impaired in magnetotaxis resulting in homogeneous cell spreading that leads to the formation of round or only slightly elongated swarm halos in semisolid agar in the presence of a magnetic field. In contrast, strains with coherent magnetosome chains and larger crystals show strong magnetic alignments (WT and $\Delta F3::mmsF$) resulting in swarm halos strongly elongated in the direction of the magnetic field (Fig. 2f, g).” (p.3, lines 118-123)

We additionally included a brief description how the swim halo aspect ratio is determined in the figure legend of Fig. 2f:

“Swim halo aspect ratio (N-to-S diameter/W-to-E diameter) within a homogenous 600 μ T magnetic field 24 h after inoculation.” (p.23, line 1021-1022)

7. CcfM is an important protein that interacts with MamY and MamK and is involved in magnetosome chain organization in MSR-1, why was the interaction between CcfM and MamF-like proteins not tested?

Thank you for the comment. We did not include CcfM into our analyses for several reasons:

- 1. CcfM is not a genuine part of the magnetoskeleton (10.1128/AEM.01976-19).*
- 2. The phenotype of the *ccfM* deletion mutant is too weak (10.1073/pnas.2014659117) to explain the drastic magnetosome chain defects in the $\Delta F3$ mutant.*
- 3. Our proteomic analyses revealed no significant change in the CcfM magnetosome abundance between the WT and $\Delta F3$.*

To clarify this, we now also indicate the proteome results for CcfM within Fig. 3:

“Circles represent non-MAI encoded proteins (e.g. the cytolinker CcfM⁵⁵).” (p.45, line 1044-1045)

8. Lines 137-138, based on the images of Extended Data Fig. 3c, the localization of MamD, Mms5, MamR, MamG in $\Delta F3::mamF$ is not restored as they showed dots instead of chains. Same for $\Delta F3::mmxF$. These do not match the description in lines 137-138.

For his conclusion, the reviewer might be misled by the well-recognizable chain-restoration phenotype of the $\Delta F3::mmsF$ strain in which all tested fluorophore-labeled proteins localize in a magnetosome chain-like pattern. However, as shown in Fig. 2, within the strain $\Delta F3::mamF$ magnetosome crystal size but not chain formation is restored. The larger size of the magnetosome crystals in $\Delta F3::mamF$ leads to their magnetism-induced aggregation at random cellular locations as MamJ is not targeted to the MM. Thus, the localization of MamD-, Mms5-, MamR-, or MamG-GFP cannot be expected to show chain-like localization patterns as in $\Delta F3::mmsF$. Instead, the fluorescence signals show (mostly) only one large single dot at varying cellular locations that correspond to the large magnetosome clusters visible in TEM micrographs. Thus, the localization pattern of the labeled proteins follows the localization of magnetosomes, indicating that MamF indeed mediates their magnetosome targeting. For strain $\Delta F3::mmxF$ the TEM analyses revealed only imperfect complementation of magnetosome sizes and chain formation and a certain degree of heterogeneity within the population (Fig. 2). Nevertheless, most cells show formation of at least short magnetosome chains at midcell. In agreement with these results, in most cells the localization of the fluorophore-labeled proteins can also be observed in short chain-like patterns at midcell within $\Delta F3::mmxF$. This localization is again reminiscent of the magnetosome localization in $\Delta F3::mmxF$ and thus justifying the conclusion that MmxF mediates targeting of the tested proteins especially in the light of the stark difference to the localization patterns observed in $\Delta F3$.

To aid in a clearer understanding of the phenotypes of the background strains and the anticipated localization patterns of the tested fluorescent proteins, we have included schematic phenotypic models in Supplementary Fig. 3c and highlighted cells with midcell chain-like fluorescence patterns in $\Delta F3::mmxF$ with arrows.

9. Lines 206-207. I am not convinced that MamJ inserts into the magnetosome membrane. How could a soluble protein insert into a membrane lipid bilayer?

We do understand the skepticism as MamJ is perfectly soluble when expressed in the absence of MFPs (Supplementary Fig. 4a; 10.1038/nature04382). However, our experiments indicate that MamJ membrane binding is independent of protein-protein interactions as magnetosome-bound MamJ is resistant against all tested extraction treatments, similar to equally treated integral membrane proteins of mitochondria (10.1038/s41467-017-00359-0). Our results are also supported by previous findings which showed that MM-bound MamJ is resistant against strong mechanical forces of at least 25 pN (10.1021/nl5017267, 10.1038/s41598-019-55804-5) as well as the lack of further mutants that also show MamJ mislocalization despite extensive targeted or random transposon mutagenesis screens (10.1128/JB.01716-14, 10.1186/s12866-021-02124-2, 10.1128/mSystems.00565-20, 10.1128/msystems.01037-21).

Nonetheless, to address the reviewer's concern, we have used cautious wording for our conclusions throughout the manuscript:

“Collectively, these results indicate that magnetosome-bound MamJ behaves as an integral membrane protein.” (p.5, lines 188-189)

Additionally, we added a section to the discussion that addresses the question raised by the reviewer: “Based on this supposed Tic20 function and the ability of MFPs to assemble into larger oligomers¹⁴, it is tempting to speculate that MFPs might form channels that facilitate the assembly of magnetosomes by inserting proteins into the MM that are not recognized by generic membrane protein integrases (e.g. the Sec translocon). Such a function could explain why MamJ or MamD have a cytosolic localization in the absence of MFPs but require detergents for their extraction from the MM when MFPs are present. The observed physical interaction between MFPs and the IMH-containing C-terminal domains of their substrates may thus only be required during the initial steps of MM insertion which would also explain why only a minor fraction of MamJ is found in complex with MmsF (Fig. 4a,b and Supplementary Fig. 4e).” (p.7, lines 306-314)

10. For Figure 1c, MamF is not really functional for protein targeting of MamD and MamJ, why not compare MmsF and MmxF with Tic20 instead of MamF?

We now provide a superimposition of MmsF with Tic20 within Fig. 1c. Notably, structural superimpositions of Tic20 with all individual MFPs revealed no significant structural changes among these proteins. To highlight the similarity of the different superimpositions, we added the following sentence to the figure legend of Fig. 1c and included the corresponding RMSD values within Supplementary Data Table S2: “Superimposition of MamF or MmxF with Tic20 yielded almost identical results (RMSDs are listed in Supplementary Data Table 2).” (p.23, lines 988-989)

And why does MamF have different functions than MmsF and MmxF?

Based on the results of the structural superimpositions, there are no major structural changes between MamF and MmsF/MmxF that could immediately explain their different functions. Therefore, further studies are required to answer this important question.

*In this context, we want to point out that similar observations could be made with Tic20, which also frequently appears in multiple copies per genome (10.4161/psb.6.7.15631) that are also often only partially functionally equivalent. The *A. thaliana* tic20-I mutant, for example, could only be complemented by Tic20-I and partially by Tic20-IV, but not by more distantly related Tic20-II and Tic20-V paralogs (10.1111/j.1365-313X.2011.04551.x). While it has been suggested that the different Tic20 paralogs are differentially expressed in different plant organs, a mechanistic understanding is still lacking (e.g. 10.1186/1471-2229-11-133).*

Minor points:

1. Line 31, to be consistent with other publications and easier to understand, please leave the strain name as “MSR-1” instead of “MSR”.

As recommended by the reviewer, the strain designation has been changed to “MSR-1” throughout the manuscript.

2. Fig. 1c, the letter “C” should be placed in the correct position.

*Thank you for noting the mispositioning of the label. We have corrected its position in **Fig. 1**.*

To make it easier for readers, please indicate that gray stands for Tic20 and purple for MamF.

*To simplify protein identification, we now included a graphic legend within **Fig. 1c** (in addition to the description within the figure legend provided in the initial submission).*

3. Fig. 3a requires a loading control to show the even loading of the samples.

*Due to their size, the loading controls were already provided in the source data file in our initial submission. We have now referenced these loading controls within the legend of **Fig. 3**: “Loading controls can be found in the source data file.” (p.23, lines 1031-1032)*

4. Fig. 4d, the TEM images are too small and the magnetosomes need to be pointed out.

*We agree with the reviewer’s comment. In **Fig. 4d** we increased the size of the TEM images and added an additional set of images with magnifications of the magnetosomes.*

5. Extended Data Figure 4B, the magnetosomes need to be pointed out.

*We agree with the reviewer’s comment. In **Fig. 4b** we increased the size of the TEM images in and added insets with magnified magnetosomes.*

Reviewer #5 (Remarks to the Author):

In this article by Paulus and colleagues, it is studied the function of the magnetosome MamF-like proteins (MFPs). They initially performed a bioinformatic study to show that these proteins belong to the HOTT family and are distantly related to Tic20 family that mediate organelle-specific protein targeting in cyanobacteria and eukaryotes. Molecular and genetic approaches allowed them to show that the MFPs are essential for the regulation of magnetosome crystal size and chain formation. However, by studying the other magnetosome proteins interacting with MFPs such as MamJ, they show that MFPs indirectly regulate organelle positioning and magnetosome crystal size by mediating correct organelle assembly.

The study is very interesting not only for the community working on biomineralization and magnetotactic bacteria but also extend to researcher working in prokaryotic and eukaryotic organelles. Thus this justify the publication of such study in a generalist journal. I have only very minor comments as I found the study very well performed with all required controls carried out.

Line 57, I would remove « Recently » as the paper cited is from 2018.

*Thanks. We changed the wording into:
“Within a previous study,...” (p. 2, line 67)*

Line 161, remove « indeed »

Agreed. We have deleted the word “indeed”. (p. 4, line 174)

please find our point-by-point responses to the reviewers' comments below with original comments/remarks in black and our answers in blue fonts:

Reviewer #2 (Remarks to the Author):

Minor comments:

— #1 Page 2 / Line 56, the acronym MTB is used, but not defined.

Thank you for pointing this out. We exchanged the acronym against: "magnetotactic bacteria" (p. 2 l. 56).

#2 The author's response to our previous comments #6.c and 6.d could be summarized and implemented in discussion.

— *Following the suggestion of reviewer 2, we added the following paragraph concerning the magnetotaxis experiments to the discussion: "Here, we further demonstrated that the ability of magnetobacterial cells to align with an external magnetic field mainly depends on their capacity to form magnetosome chains. Swarm halos elongated in the direction of the magnetic field were exclusively observed in strains capable of at least rudimentary magnetosome chain formation (e.g. $\Delta F3::mmsF$). In contrast, magnetosome chain-free strains were unable to align in the magnetic field and formed only equidimensional swarm halos ($\Delta F3$, $\Delta F3::mamF$, and $\Delta mmsF/mmsF$). Notably, we also observed an unexpected difference between the isogenic strains $\Delta F3::mmsF$ and $\Delta mamF/mmsF$. Although both strains express *mmsF* as the sole MFP, only $\Delta F3::mmsF$ showed a wildtype-like magnetotactic behavior. We attribute this difference to the stronger expression of *mmsF* in strain $\Delta F3::mmsF$ due to the use of a stronger promoter (*P_{mamG}* vs. the natural *P_{mms6}* in strain $\Delta mamF/mmsF$)²⁶. Elevated *mmsF* expression in $\Delta F3::mmsF$ likely improves the targeting of biomineralization proteins such as MamD and Mms5, resulting in larger, more magnetic crystals and enhanced cohesion within the magnetosome chains (Fig. 2a, c, e). This increased cohesion, accompanied by a reduced number of "free" magnetosomes, is likely due to enhanced recruitment of MamJ (Fig. 3a). The swarm agar assay may thus be sensitive enough to quantify the relative contributions of magnetosome chain formation and magnetite crystal size for magnetotaxis. Such experiments, however, would require strictly controlled conditions due to the growth-dependent establishment of nutrient and oxygen gradients that caused slight differences in formation of elongated swarm halos over time during our experiments (Fig. 2f, g)." (page 7-8, ll. 312-330).*

#3

In response to our previous comment #8, the authors updated the figure with a new TEM micrograph to emphasize their statement. For transparency reasons, the authors could add the previous Fig. 2b, e panels in the Supplement to indicate some variation.

*Thank you for the suggestion. We included the $\Delta F3::mmsF$ image from our initial submission within Supplementary Figure S2d: “d Expression of *mmsF* in $\Delta F3$ results in phenotypic heterogeneity (compare with $\Delta F3::mmsF$ TEM image in Fig. 2c) which may derive from an unbalanced production of MmsF or an impaired folding due to the use of the strong *mamG* promoter.” (Supplementary information, Supplementary figure 2 legend).*

*We therefore have also added the following sentence to the manuscript to refer to the new image: “Individual expression of *mmsF* or *mamF* within $\Delta F3$ restored magnetosome crystal sizes to the level of double deletion mutants whereas complementation with *mmsF* restored crystal sizes incompletely with a certain degree of phenotypic heterogeneity as indicated by the bimodal crystal size distribution (Fig. 2a, c and Supplementary Fig. 2d).” (p. 3, ll. 107-110).*

#4

In their reply to our previous comment #9a, about the increase of MamK in the magnetosome fraction, the authors suggest that there might be another binding partner. This might be an interesting point for the discussion.

Thank you for the suggestion. We added a small paragraph to the discussion to highlight the finding that MamK still binds to magnetosomes in the almost complete absence of MamJ: “MamJ is crucial for magnetosome chain formation by tethering magnetosomes to the cytosolic MamK filament (Supplementary Fig. 3a) ¹⁵. Given the almost complete absence of MamJ from the magnetosome membrane of strain $\Delta F3$, we anticipated a significant reduction in MamK binding to magnetosomes. Surprisingly, however, proteomic and immunoblot analyses revealed wildtype-like MamK levels on the magnetosome membrane of $\Delta F3$. This unexpected finding suggests that MamK interacts with additional magnetosome proteins beyond MamJ. However, no such interactions have been reported to date. We thus hypothesize that a protein with unaffected or slightly elevated abundance in the $\Delta F3$ MM may also have the capacity to bind MamK, albeit weakly, as evidenced by the lack of chain formation in this strain. Further investigations will be necessary to identify such potential MamK-binding partners and clarify their roles in magnetosome organization.” (p. 6, ll. 278-288).

Reviewer #3 (Remarks to the Author):

We greatly appreciate your effort and valuable contributions in co-reviewing this manuscript.

Reviewer #4 (Remarks to the Author):

The authors revised well to the comments and made the necessary corrections to the text and figures.

Thank you very much for reviewing this manuscript.